# Rational Water and Nitrogen Regulation Can Improve Yield and Water–Nitrogen Productivity of the Maize (*Zea mays* L.)–Soybean (*Glycine max* L. Merr.) Strip Intercropping System in the China Hexi Oasis Irrigation Area

**DOI:** 10.3390/plants14132050

**Published:** 2025-07-04

**Authors:** Haoliang Deng, Xiaofan Pan, Guang Li, Qinli Wang, Rang Xiao

**Affiliations:** 1College of Civil Engineering, Hexi University, Zhangye 734000, China; denghaoliang521@163.com (H.D.); panxiaofan2023@163.com (X.P.); xiaorang999@163.com (R.X.); 2Gansu Provincial Engineering Research Center for the Resource Utilization of Edible Fungi and Fungi Bran, Hexi University, Zhangye 734000, China; wangqinli66@163.com; 3College of Agriculture and Ecological Engineering, Hexi University, Zhangye 734000, China

**Keywords:** maize–soybean strip intercropping, water–nitrogen coupling, quality, yield, comprehensive growth model

## Abstract

The planting area of the maize–soybean strip intercropping system has been increasing annually in the Hexi Corridor oasis irrigation area of China. However, long-term irrational water resource utilization and the excessive mono-application of fertilizers have led to significantly low water and nitrogen use efficiency in this cropping system. To explore the sustainable production model of high yield and high water–nitrogen productivity in maize–soybean strip intercropping, we established three irrigation levels (low: 60%, medium: 80%, and sufficient: 100% of reference crop evapotranspiration) and three nitrogen application levels (low: maize 230 kg ha^−1^, soybean 29 kg ha^−1^; medium: maize 340 kg ha^−1^, soybean 57 kg ha^−1^; and high: maize 450 kg ha^−1^, soybean 85 kg ha^−1^) for maize and soybean, respectively. Three irrigation levels without nitrogen application served as controls. The effects of different water–nitrogen combinations on multiple indicators of the maize–soybean strip intercropping system, including yield, water–nitrogen productivity, and quality, were analyzed. The results showed that the irrigation amount and nitrogen application rate significantly affected the kernel quality of maize. Specifically, the medium nitrogen and sufficient water (N2W3) combination achieved optimal performance in crude fat, starch, and bulk density. However, excessive irrigation and nitrogen application led to a reduction in the content of lysine and crude protein in maize, as well as crude fat and crude starch in soybean. Appropriate irrigation and nitrogen application significantly increased the yield in the maize–soybean strip intercropping system, in which the N2W3 treatment had the highest yield, with maize and soybean yields reaching 14007.02 and 2025.39 kg ha^−1^, respectively, which increased by 2.52% to 138.85% and 5.37% to 191.44% compared with the other treatments. Taking into account the growing environment of the oasis agricultural area in the Hexi Corridor and the effects of different water and nitrogen supplies on the yield, water–nitrogen productivity, and kernel quality of maize and soybeans in the strip intercropping system, the highest target yield can be achieved when the irrigation quotas for maize and soybeans are set at 100% ET0 (reference crop evapotranspiration), with nitrogen application rates of 354.78~422.51 kg ha^−1^ and 60.27~71.81 kg ha^−1^, respectively. This provides guidance for enhancing yield and quality in maize–soybean strip intercropping in the oasis agricultural area of the Hexi Corridor, achieving the dual objectives of high yield and superior quality.

## 1. Introduction

The rapid growth of the global population has not only placed tremendous pressure on food production, but has also caused severe impacts on ecological environments and sustainable agricultural development [1]. Diversified cropping systems, such as intercropping, represent a viable strategy for sustainable agricultural development and are one of the most important measures to maintain crop yields and ensure food security in the current fragile agricultural ecological environment [2,3]. Compared with monoculture systems, intercropping legumes (Leguminosae) and cereals (Gramineae) creates reasonable interspecific interaction conditions, leading to significant mutual benefits in nutrient uptake and yield performance, thereby demonstrating distinct productivity advantages [4]. Therefore, in traditional agricultural practices, particularly in the oasis agricultural area of the Hexi Corridor in China, this technique has been widely adopted to address continuous cropping obstacles, increase the multiple cropping index, and ultimately enhance crop yield and economic benefits. However, traditional intercropping systems face multiple challenges, including irrational field arrangements, difficulties in crop rotation, intense competition for land resources, and a lack of synchronized fertilization and pest and weed control techniques. These issues result in unstable and low yields, ultimately constraining sustainable development [5]. The maize–soybean strip intercropping planting system is an upgrade of traditional intercropping and can ensure no reductions in the yield of maize and an increased harvest of soybeans [6]. The staggered arrangement of maize and soybean in height can effectively maximize crop utilization of light energy and arable land, promote crop synergy, double the harvest, improve systematic crop yields, and effectively alleviate the competition for land by grain and oil. It is an effective strategy to achieve sustainable ecological agricultural development and the revitalization of the soybean industry [5]. In recent years, driven by the pursuit of higher crop yields, excessive irrigation and fertilizer application by farmers have led to a series of environmental issues, including water loss, inefficient fertilizer use, and pollution of agricultural ecosystems [7,8]. Moreover, these practices adversely affect farmland soil quality, crop root development, and the balanced improvement of yield and quality. Rational water–fertilizer coupling techniques can enhance rapid nutrient uptake and efficient utilization by crop roots, achieving the dual objectives of “regulating water through fertilization” and “promoting fertilizer efficiency via water management.” This approach ultimately realizes water and fertilizer conservation while simultaneously increasing yield and improving quality [9]. The effects of nitrogen application on soybeans may vary significantly depending on regional climatic conditions, soil fertility, and cultivation management measures. In Minnesota, soybeans can meet their nitrogen requirements through rhizobial nitrogen fixation and soil mineralization, making the yield-increasing effect of additional nitrogen application limited [10]. In contrast, in semi-arid and semi-humid regions of China, rational increases in nitrogen application can enhance yield and water use efficiency by improving leaf photosynthetic performance [11]. Therefore, the management of nitrogen for soybeans should adhere to the regionalization principle, with fertilization strategies tailored to the specific characteristics of the local soil–climate–crop system to achieve synergistic improvements in both yield and nitrogen use efficiency.

The supply of water and fertilizer not only affects the crop yield and water–fertilizer use efficiency, but also influences fruit quality. A study by Ma et al. [12] demonstrated that rational water–nitrogen management can promote the translocation of photosynthetic assimilates to fruits in wolfberry plants, increasing yield by 3.45%~21.53% while improving fruit quality, thereby achieving both high yield and superior quality. Fan et al. [13] revealed that appropriately reducing water supply, combined with nitrogen application, could increase tomato yield and positively regulate fruit quality, albeit at the cost of reduced water–nitrogen use efficiency. Research conducted by Guerra et al. [14] demonstrated that deficit irrigation, combined with nitrogen application, could increase vitamin C content and juice concentration while reducing acidity in red tangerines, while simultaneously maintaining high yield and economic benefits. In the strip intercropping system, scientifically regulating the irrigation amount and nitrogen application rate based on the growth and development patterns of maize and soybeans can effectively enhance water and nitrogen use efficiency, increase yield, and improve grain quality. This approach holds significant scientific importance for promoting high and stable yields, as well as quality improvement of maize and soybeans in the Hexi Corridor area of China.

The effects of different water and nitrogen application rates on crop yield, quality, and water–nitrogen use efficiency vary significantly. Previous studies on water and nitrogen management in maize–soybean strip intercropping system have primarily focused on analyzing the variation patterns of growth indicators through field experiments. However, the comprehensive evaluation of yield and quality remains in the initial exploration stage. Although some researchers have adapted principal component analysis, gray relational analysis, and cluster analysis methods to conduct comprehensive evaluations of indicators such as kernel yield and quality, these algorithms have limitations. They feature single evaluation objectives and fail to precisely identify the optimal water–nitrogen coupling thresholds that simultaneously achieve superior solutions in multiple dimensions, including yield, quality, and water–nitrogen use efficiency. Consequently, a comprehensive consideration of multiple indicators’ responses to water–nitrogen management patterns is required. This necessitates upgrading and optimizing the multi-objective evaluation system to derive holistic optimization solutions. The combined weighting approach based on game theory effectively integrates the weights derived from both the AHP method and the entropy weight method, overcoming the bias inherent in single evaluation methods. This hybrid approach enables quantitative analysis of qualitative problems, yielding more reasonable indicator weights. By employing the TOPSIS method to construct the evaluation system, it achieves the scientific assessment of multi-objective criteria [15]. Furthermore, a binary equation was introduced to conduct a secondary analysis of the TOPSIS evaluation system, which explicitly quantified the single factor and interactive effect of water and nitrogen inputs on the evaluation system. This approach delineated the threshold of water–nitrogen coupling, thereby achieving highly efficient utilization of water and nitrogen resources. This study employs a maize–soybean strip intercropping system as the experimental platform, conducting a factorial experiment with four nitrogen application rates and three irrigation levels. By adopting a comprehensive multi-indicator evaluation system, we have examined the interactive effects of water and nitrogen on yield components, kernel yield, water–nitrogen productivity, and quality in the maize–soybean strip intercropping system; established a comprehensive multi-indicator evaluation system; built water–nitrogen comprehensive evaluation regression models; and proposed optimal water–nitrogen combination schemes under multi-objective synergy. This will provide theoretical foundations for implementing integrated water–nitrogen management strategies, achieving sustainable production with high yield and superior quality and improving water–nitrogen productivity in the maize–soybean strip intercropping system.

## 2. Materials and Methods

### 2.1. Experimental Site Profile

The experiment was conducted from April 2023 to October 2024 at Yimin Irrigation Experimental Station, Minle County, the middle part of the Hexi Corridor, Gansu Province (100°43′ E, 38°39′ N, 1970 m a.s.l.) (Figure 1). The climate in the experimental area is temperate continental. The average annual precipitation is approximately 200 mm, the evaporation is 1900 mm, the average annual temperature is 6.0 °C, and the frost-free period lasts about 105 d. The experimental soil is light loam, with a maximum field water-holding capacity of 24% in the tillage layer and a bulk density of 1.46 g cm^−3^. The 0–20 cm soil layer has a pH of 7.7 and an organic matter content of 11.9 g kg^−1^. The available phosphorus, available potassium, and alkali-hydrolyzable nitrogen contents are 17.3 mg kg^−1^, 174.9 mg kg^−1^, and 63.1 mg kg^−1^, respectively, indicating moderate fertility. The precipitation and the average temperature over the experiments are shown in Figure 2.

### 2.2. Experimental Design

The experiment used “*Longdan 6*” maize and “*Zhonghuang 30*” soybean as the test crops, with three irrigation levels and four nitrogen application levels. The irrigation levels were determined based on the reference crop evapotranspiration, which was calculated using the Penman–Monteith equation [16]. The crop coefficient values for the different growth stages of maize and soybean were taken from references [17,18]. The nitrogen application rates were determined according to the conventional nitrogen application rates used by local farmers in maize and soybean fields near the experimental area. The three irrigation levels were set as follows: low water (W1, 60% ET_0_), medium water (W2, 80% ET_0_), and sufficient water (W3, 100% ET_0_). The four nitrogen application levels were as follows: no nitrogen (N0, 0 kg ha^−1^), low nitrogen (N1, maize: 230 kg ha^−1^, soybean: 29 kg ha^−1^), medium nitrogen (N2, maize: 340 kg ha^−1^, soybean: 57 kg ha^−1^), and high nitrogen (N3, maize: 450 kg ha^−1^, soybean: 85 kg ha^−1^). The no-nitrogen treatment (N0) served as the control, resulting in a total of 12 treatments, with each treatment being replicated three times. The experimental plots were arranged in a randomized block design, with each plot measuring 20 m in length and 6 m in width (120 m^2^ total area). To prevent water and nutrient interactions between adjacent plots, 1.5 m deep PVC plastic barriers were installed along the plot boundaries, and 20 cm high ridges were constructed around each plot.

The experiment adopted drip irrigation under plastic mulch technology, with independent water and fertilizer supply pipelines arranged separately for maize and soybean. The sowing and harvesting schedule was as follows: in 2023, maize and soybean were sown on 21 April, soybean was harvested on 7 September, and maize was harvested on 30 September; in 2024, maize and soybean were sown on 16 April, soybean was harvested on September 1, and maize was harvested on 28 September. Based on the preliminary experimental results, this study adopted a 2:3 maize–soybean strip intercropping system, consisting of two rows of maize alternated with three rows of soybean (Appendix A). The planting configuration was as follows: the maize row spacing was 0.4 m, the plant spacing was 0.11 m, the planting density was 82,685 plants ha^−1^, the soybean row spacing was 0.3 m, the plant spacing was 0.10 m, and the planting density was 90,954 plants ha^−1^. The spacing between the maize and soybean rows was maintained at 0.6 m. Each experimental plot was equipped with an individual water meter for precise irrigation control. The application rates of phosphate (diammonium phosphate, P_2_O_5_ ≥ 46%) and potassium (potassium sulfate, K_2_O ≥ 52%) were the same in all treatments of maize, which were 150 and 120 kg ha^−1^, respectively. The nitrogen fertilizer was applied four times at different growth stages, which were 20% total nitrogen at the basal fertilizer, 30% total nitrogen at the jointing stage, 30% total nitrogen at the tasseling stage, and 20% total nitrogen at the filling stage, respectively. The soybean treatments had the same phosphate (diammonium phosphate, P_2_O_5_ ≥ 46%) and potassium (potassium sulfate, K_2_O ≥ 52%) application rates of 25 and 33 kg ha^−1^, respectively, and 50% of the nitrogen fertilizer was applied at the sowing stage and podding stage (Appendix A). Table 1 presents the irrigation quotas, nitrogen application rates, and normalized coded values for all maize and soybean treatments.

### 2.3. Measurement Items and Methods

#### 2.3.1. Meteorological Data

The meteorological data include precipitation, solar radiation, air temperature, relative air humidity, wind speed etc., which were collected from the microclimate monitoring station (Model MC-NQXZ, Jinzhou Sunshine Meteorological Technology Co., Ltd., Jinzhou, China) near the experimental site.

#### 2.3.2. Water Productivity

Soil moisture was monitored using TRIME-PICO TDR (IMKO Micromodultechnik GmbH, Ettlingen, GER). The measurements were conducted at 7-day intervals throughout the whole growth stage of maize and soybean, with additional measurements being taken before sowing and after harvest, as well as before and after precipitation events, 1 day before irrigation, and 2 days after irrigation. The sampling depth was 100 cm, with an interval of 20 cm. Crop water consumption (ET) was calculated using the water balance method. Given the adoption of water-saving drip irrigation under plastic mulch, evaporation losses were negligible (i.e., infiltration efficiency of irrigation reached 100%). For rainfall infiltration, the actual infiltration amount under mulching conditions was determined by calculating the increase in soil water content before and after each rainfall event:(1)W=∑d×ρ×ω×10
where *W* is soil water storage, mm; *d* is soil layer depth, mm; *ρ* is bulk density at depth h, g cm^−3^; and *ω* is soil water content, %.

The water consumption was calculated using the water balance equation, expressed as:(2)$$ET=\sum_{i=1}^{n}\left(W_{if}-W_{ib}\right)+I+\Delta W-Q$$
where *ET* is total crop water consumption during the whole growth stage (mm); *W_if_* is soil water storage in the 100 cm soil layer after the *i*-th rainfall event (mm); *W_ib_* is soil water storage in the 100 cm soil layer before the *i*-th rainfall event (mm); *I* is the effective irrigation amount (mm); Δ*W* is the change in soil water storage (mm); and *Q* is groundwater recharge and leakage (mm).

As the groundwater table depth in the experimental area exceeded 10 m, the irrigation method was drip-irrigated, and no runoff drainage occurred during the growth stage. Therefore, groundwater recharge and leakage were neglected in the calculations:(3)$$\mathrm{Water}\;\mathrm{productivity}\;(\mathrm{WP},\;\mathrm{kg}\;m^{-3})=Y/\mathrm{ET}/10$$(4)$$\mathrm{Irrigation}\;\mathrm{water}\;\mathrm{productivity}\;(\mathrm{IP},\;\mathrm{kg}\;m^{-3})=Y/I/10$$
where *Y* is maize yield (kg ha^−1^) [19].

#### 2.3.3. Nitrogen Use Efficiency (NUE)

(5)$$\mathrm{Nitrogen}\;\mathrm{partial}\;\mathrm{factor}\;\mathrm{productivity}\;(\mathrm{NPF},\;\mathrm{kg}\;{\mathrm{kg}}^{-1})=Y/N$$
where *N* is nitrogen fertilizer input (kg ha^−1^) [19].(6)$$\mathrm{Nitrogen}\;\mathrm{agronomic}\;\mathrm{use}\;\mathrm{efficiency}\;(\mathrm{NAE},\;\mathrm{kg}\;{\mathrm{kg}}^{-1})=(Y_{N}-Y_{0})/N$$
where *Y_N_* is grain yield in a nitrogen-fertilized plot (kg ha^−1^) and *Y_0_* is grain yield in a zero-nitrogen control plot (kg ha^−1^) [19].

#### 2.3.4. Above-Ground Dry Matter

During the maturity stage of maize and soybeans, 10 standard sample plants were randomly selected from non-border rows in each experimental plot. The above-ground parts were bagged, deactivated in a constant-temperature oven at 105 °C for 30 min, and then dried at a constant temperature of 80 °C until a constant weight was achieved. The samples were weighed using an electronic balance with a precision of 0.01 g (the brand is Ji Ming glass high windshield electronic balance; the serial number is JM2003B+, Dongguan Mingke Instrument Co., Ltd., Dongguan City, Guangdong Province, China.), and the average value was calculated. 

#### 2.3.5. Yield

After the maize and soybeans reached maturity, 30 standard sample plants were randomly selected from non-border rows in each experimental plot. These plants were brought back to the laboratory, where yield component indicators were measured after natural air-drying. The remaining plants were harvested separately and threshed after air-drying to determine grain yield, which was then converted to yield per hectare.

#### 2.3.6. Quality

The crude fat content was determined using the Soxhlet extraction method (GB 5009.6-2016) [20]. The process involved extraction using petroleum ether at 60–90 °C, boiling for 20 min at a boiling temperature of 125 °C, rinsing for 45 min, extracting for 8 h, drying at 100 °C for 30 min, cooling in a desiccator for 45 min, and then weighing until a constant weight was achieved. The crude protein content was measured by the semi-micro Kjeldahl method (GB 5009.5-2016) [21], where the temperature of the digestion furnace reached 420 °C and the digestion was continued for 1 h. After cooling, 50 mL of water was added, and the process of automated reagent addition, distillation, titration, and titration data recording was carried out using an automatic Kjeldahl nitrogen analyzer, where the concentration of the standard hydrochloric acid titration solution was 1.0 mol L^−1^. The starch content was analyzed using the acid hydrolysis-DNS method (GB 5009.9-2016) [22], fats were washed off with 50 mL of petroleum ether, and the residue was rinsed with 150 mL of 85% ethanol. Then, 30 mL of hydrochloric acid was added, and the mixture was heated in a 100 °C water bath for 30 min, followed by refluxing for 2 h. After cooling, 2 drops of methyl red indicator solution were added, and titration was performed using 400 g L^−1^ sodium hydroxide solution with hydrochloric acid correction. Next, 20 mL of 200 g L^−1^ lead acetate solution was added, and the mixture was allowed to stand for 10 min. Subsequently, 20 mL of 100 g L^−1^ sodium sulfate solution was added. Finally, the absorbance of the filtrate was measured at 540 nm. The lysine content was determined by the ninhydrin colorimetric method (GB 5009.124-2016) [23], where 10–15 mL of 6 mol L^−1^ hydrochloric acid solution and 3–4 drops of phenol were added into the hydrolysis tube. The mixture was frozen for 3–5 min, evacuated under vacuum, and then purged with nitrogen. It was hydrolyzed in a 110 °C hydrolysis oven for 22 h and then cooled to room temperature. The hydrolysate was filtered into a 50 mL volumetric flask. A 1.0 mL aliquot of the filtrate was evaporated to dryness at 40–50 °C and then dissolved in pH 2.2 sodium citrate buffer solution. The absorbance of the filtrate was measured at 570 nm. The bulk density was measured using a JC-GHCS-1000 electronic grain density tester (Qingdao Juchuang Environmental Protection Group Co., Ltd., Shandong, China) [24].

### 2.4. Data Statistics and Analysis

The statistical analysis and regression model establishment were performed using the LSD multiple comparison method in SPSS 22.0 (IBM, Inc., New York, NY, USA) software, while the graphs were plotted using Origin 8.0 (Origin Lab, Corp., Hampton, MA, USA). Yaaph (Meta Decision Software Technology Co., Ltd., Corp., Taiyuan, China) software was employed to construct the hierarchical model for comprehensive analysis of the maize–soybean intercropping system and to analyze the weights of various indicators. Matlab (Version R2023 b, MathWorks, Corp., Natick, MA, USA) was used to calculate the combined weights based on game theory and the comprehensive scores of TOPSIS, as well as to interpret the regression model.

## 3. Results and Analysis

### 3.1. Water–Nitrogen Interaction Effects on Yield and Dry Matter Accumulation of Maize and Soybean in the Strip Intercropping System

#### 3.1.1. Water–Nitrogen Interaction Effects on Maize Yield and Dry Matter Accumulation in the Strip Intercropping System

As shown in Figure 3, the maize yield and dry matter accumulation followed consistent patterns in response to water and nitrogen inputs during the two-year experimental period, both exhibiting a rapid initial increase followed by a plateau. Variance analysis revealed that, in the Hexi Oasis irrigation area, the irrigation amount was the primary factor affecting maize yield, while the nitrogen application rate was the dominant factor influencing maize dry matter accumulation. An analysis of the two-year averaged data revealed that maize yield and dry matter accumulation showed an increasing trend with elevated irrigation or nitrogen application rates under the same nitrogen or irrigation levels, respectively. When averaging the treatments with the same irrigation amount, the results demonstrated that yield and dry matter accumulation increased by 36.79% and 14.01%, respectively, from low to medium irrigation levels, and by 10.81% and 6.39% from medium to sufficient irrigation levels, respectively. This demonstrates that the yield-enhancing effect of maize gradually diminishes as the irrigation volume increases. The nitrogen application rate also had a significant effect on maize yield. Compared to that observed with no nitrogen application, low nitrogen application increased the yield by 35.56%, and medium nitrogen application further increased it by 20.45% compared to low nitrogen application. However, when the nitrogen application rate increased to the N3 level, there was no significant change in yield, which only increased by 4.72%. The trend in dry matter accumulation was consistent with the yield performance. After analyzing the water–nitrogen interaction, it was found that the combination of medium nitrogen and sufficient water levels (N2W3) achieved the highest yield and dry matter mass, reaching 14,007.02 kg ha^−1^ and 35,134.30 kg ha^−1^, respectively. This indicates that sufficient irrigation can lead to high yields, but either insufficient or excessive nitrogen application can inhibit maize growth and limit yield increase.

#### 3.1.2. Water and Nitrogen Interaction Effects on Soybean Yield and Dry Matter Accumulation in the Strip Intercropping System

As shown in Figure 4, the variation trends of soybean yield and dry matter accumulation were generally consistent with those of maize. The analysis of variance (ANOVA) revealed that the nitrogen application rate and irrigation amount had a greater effect on soybean yield than on dry matter accumulation. When averaged across the same nitrogen application rate, the results showed that, from no to low nitrogen application, the yield and dry matter accumulation increased by 49.00% and 64.09%, respectively; from low to medium nitrogen levels, the yield and dry matter increased by 31.01% and 43.61%, respectively; and, when the nitrogen application rate exceeded 140 kg ha^−1^, the yield only increased by 2.94%, while the dry matter accumulation still showed a significant increase of 11.05%. The amount of irrigation also had a significant impact on soybean yield and dry matter accumulation. From low to medium irrigation levels, and from medium to sufficient irrigation levels, the yield and dry matter accumulation increased by 39.82% and 54.13%, and 10.46% and 23.13%, respectively. Under the interaction of water and nitrogen, the optimal combination for yield and dry matter accumulation was medium nitrogen and sufficient water levels (N2W3), achieving 2024.89 kg ha^−1^ and 5431.84 kg ha^−1^, respectively. This indicates that, under sufficient irrigation conditions, the amount of nitrogen application determines the dry matter accumulation and yield of soybeans. Insufficient or excessive nitrogen application will result in a decrease in yield.

### 3.2. Water–Nitrogen Interaction Effects on Yield Components of Maize and Soybean in the Strip Intercropping System

#### 3.2.1. Water–Nitrogen Interaction Effects on Maize Yield Components in the Strip Intercropping System

As shown in Table 2, the nitrogen application rate was the primary factor affecting the maize yield components. The averaged data across equivalent nitrogen levels revealed that, from no to low nitrogen treatment, the ear length, ear diameter, kernel number per ear, and ear weight increased by 11.01%, 6.80%, 16.47%, and 16.24%, respectively; moreover, from low to medium nitrogen levels, these parameters further increased by 7.16%, 3.20%, 10.93%, and 20.90%, respectively. When the nitrogen application rate exceeded 340 kg ha^−1^, the increased in ear length, ear diameter, kernel number per ear, and ear weight were merely 1.21%, 0.46%, 1.91%, and 3.80%, respectively, indicating that excessive nitrogen fertilization did not demonstrate a sustained improvement effect on the yield components.

#### 3.2.2. Water–Nitrogen Interaction Effects on Soybean Yield Components in the Strip Intercropping System

As shown in Table 3, the transition from no to low nitrogen application significantly enhanced soybean yield components, with the number of effective pods per plant increasing by 25.18%, the kernel weight per plant rising by 44.11%, the kernel number per plant growing by 28.92%, and the 100-kernel weight improving by 30.91%. When the nitrogen application rate increased from low to medium, these parameters increased by 10.68%, 26.00%, 11.69%, and 19.04%, respectively. However, exceeding the threshold of 140 kg ha^−1^ nitrogen application, and further increases in nitrogen fertilizer application, had no significant effect on the yield components. The yield components of soybeans were also influenced by the irrigation amount. The averaged data across equivalent irrigation levels revealed that, from low to medium water levels, the number of effective pods per plant, the kernel weight per plant, the kernel number per plant, and the 100-kernel weight increased by 18.96%, 29.40%, 17.84%, and 28.02%, respectively. From medium to sufficient water levels, these parameters increased by 5.89%, 16.45%, 6.62%, and 10.24%, respectively.

### 3.3. Water–Nitrogen Interaction Effects on WP of Maize and Soybean in the Strip Intercropping System

#### 3.3.1. Water–Nitrogen Interaction Effects on Maize WP in the Strip Intercropping System

As shown in Table 4, under the same nitrogen application level, water consumption increased with an increasing irrigation amount, rising by 20.67% from low to medium water levels and by 17.1% from medium to sufficient water levels. The application of additional nitrogen fertilizer could promote rapid maize growth and enhance the water uptake rate by the roots. The averaged data across equivalent nitrogen application rates revealed that water consumption increased by 14.27% from no nitrogen to low nitrogen application, by 5.24% from low nitrogen to medium nitrogen application, and by 7.55% from medium nitrogen to high nitrogen application. WP and IP were predominantly influenced by the nitrogen application rates. Compared to no nitrogen application, low nitrogen application increased these metrics by 18.30% and 34.34%, respectively. Further increases from low to medium nitrogen levels resulted in improvements of 13.78% and 19.06%, respectively, while the progression from medium to high nitrogen levels yielded smaller gains of 0.21% and 8.65%, respectively. Under the interaction of water and nitrogen application, the high nitrogen and medium water (N3W2) treatment achieved the highest WP and irrigation WP.

#### 3.3.2. Water–Nitrogen Interaction Effects on Soybean WP in the Strip Intercropping System

As shown in Table 4, from low to sufficient irrigation levels, water consumption increased by 25.81%~43.69%, while, from no nitrogen to high nitrogen application levels, water consumption increased by 8.52%~22.02%. Nitrogen application promoted soybean plant growth and enhanced root water absorption capacity. From no to medium nitrogen application, soybean WP and IP increased by 17.70%~21.95% and 29.81%~47.91%, respectively. When increasing from medium to high nitrogen levels, IP rose by 4.37%, while WP decreased by 5.31%. Similarly, the irrigation amount also affected soybean WP and IP. Specifically, when increasing from low to medium irrigation levels, IP improved by 4.85%. However, as irrigation increased to sufficient levels, IP decreased by 11.61%. Meanwhile, WP showed a continuous declining trend from low to sufficient irrigation levels, with a reduction range of 4.03%~12.16%. Under water–nitrogen interaction, medium nitrogen and low water levels (N2W1) achieved the highest WP, while high nitrogen and medium water levels (N3W2) attained the maximum IP.

### 3.4. Water–Nitrogen Interaction Effects on NUE of Maize and Soybean in the Strip Intercropping System

#### 3.4.1. Water–Nitrogen Interaction Effects on NUE of Maize in the Strip Intercropping System

As shown in Table 5, under the same nitrogen application rate, both NPF and NAE in maize showed an increasing trend with the increase in irrigation. Specifically, from low to medium water levels, NPF and NAE increased by 40.33% and 104.36%, respectively; moreover, from medium to sufficient water levels, they further increased by 10.63% and 7.59%, respectively. Under the same irrigation level, from low to medium nitrogen application rates, NAE increased marginally by 3.54%, while NPF decreased by 6.13%; moreover, from medium to high nitrogen application rates, both NPF and NAE declined by 9.69% and 3.06%, respectively. Under the interaction of water and nitrogen, the highest NPF was achieved with low nitrogen and sufficient water levels (N1W3), while the highest NAE was obtained with medium nitrogen and sufficient water levels (N2W3). However, NPF and NAE did not show consistency in their response to nitrogen application rates. A lower nitrogen application rate resulted in higher NPF, whereas higher NAE required a greater amount of nitrogen application.

#### 3.4.2. Water–Nitrogen Interaction Effects on NUE of Soybean in the Strip Intercropping System

As shown in Table 5, NPF decreased by 3.85% from low to medium nitrogen levels and by 7.17% from medium to high nitrogen levels. In contrast, NAE increased by 7.09% from low to medium nitrogen levels and by 1.38% from medium to high nitrogen levels. Increasing the irrigation amount could enhance the NPF and NAE in soybeans. The averaged data across equivalent irrigation amounts revealed that, from low to medium irrigation levels, NPF and NAE increased by 45.28% and 90.12%, respectively; moreover, from medium to sufficient irrigation levels, NAE further increased by 8.23%, while NPF decreased by 4.35%. An integrated analysis of individual and interactive water–nitrogen effects reveals that the highest NAE did not occur under high nitrogen and sufficient water levels (N3W3), but rather under medium nitrogen and sufficient water levels (N2W3). Similarly, the highest NPF was not observed under low nitrogen and medium water levels (N1W2), but instead under low nitrogen and sufficient water levels (N1W3). This demonstrates that adequate irrigation can effectively enhance NPF under water–nitrogen interactions.

### 3.5. Water–Nitrogen Interaction Effects on the Quality of Maize and Soybean in the Strip Intercropping System

#### 3.5.1. Water–Nitrogen Interaction Effects on Maize Quality in the Strip Intercropping System

The analysis of the effects of water and nitrogen interaction on maize quality in the strip intercropping system over two years revealed that the irrigation amount and nitrogen application rate significantly influenced the starch, crude protein, and lysine content in maize kernels. However, the water–nitrogen interaction only had a significant impact on crude protein and lysine (Table 6). The averaged data across equivalent nitrogen application rates revealed that, from no nitrogen to low nitrogen application, the crude fat, starch, crude protein, lysine content, and bulk density of the maize kernels increased by 8.18%, 20.54%, 36.11%, 24.41%, and 1.44%, respectively. From low nitrogen to medium nitrogen application, the starch, crude protein, lysine content, and bulk density increased by 10.52%, 8.05%, 33.54%, and 1.37%, respectively, but this was unfavorable for crude fat accumulation, which decreased by 1.04%. From medium nitrogen to high nitrogen application, only the crude fat content and bulk density increased by 3.85% and 0.71%, respectively, while the starch, crude protein, and lysine content decreased by 5.05%, 14.86%, and 12.32%, respectively. The averaged data across equivalent irrigation amounts revealed that, from low to medium irrigation levels, maize kernel starch content and bulk density increased by 14.57% and 2.46%, respectively. From medium to sufficient irrigation levels, the starch content and bulk density further increased by 8.06% and 2.10%, respectively, while the crude fat, crude protein, and lysine content decreased by 1.08%, 15.35%, and 4.12%, respectively. It could be observed that the crude fat content and bulk density show a sustained increasing trend with a rising nitrogen application rate, although the rate of increase gradually slowed. In contrast, the contents of starch, crude protein, and lysine initially increased and subsequently decreased as the nitrogen application rate increased. Additionally, increasing the irrigation amount was beneficial for improving the starch content and bulk density, whereas excessive water supply led to reductions in crude fat, crude protein, and lysine contents.

#### 3.5.2. Water–Nitrogen Interaction Effects on Soybean Quality in the Strip Intercropping System

The analysis of the two-year water–nitrogen interaction effects on soybean quality in the strip intercropping system revealed that the nitrogen application rate had a significant impact on soybean crude fat content, whereas neither individual water or nitrogen factors nor their interaction significantly affected the crude protein content (Table 6). The averaged data across equivalent nitrogen application rates revealed that the crude protein content increased with higher nitrogen application rates. Specifically, from no nitrogen to low nitrogen application, there was an increase of 2.08%; from low nitrogen to medium nitrogen application, there was an increase of 2.85%; and from medium nitrogen to high nitrogen application, there was an increase of 2.13%. In contrast, the crude fat content exhibited an initial increase followed by a decreasing trend with rising nitrogen application. Specifically, from no nitrogen to low nitrogen application, there was an increase of 8.27%; from low nitrogen to medium nitrogen application, there was an increase of 3.85%; and from medium nitrogen to high nitrogen application, there was a decrease of 1.87%. From the averaged data across equivalent irrigation amounts, it was found that excessive irrigation negatively affected the accumulation of crude protein and crude fat contents. Specifically, from low to medium irrigation levels, crude protein decreased by 6.20%; moreover, although crude protein increased by 1.50% from medium to sufficient irrigation levels, it remained lower than the low irrigation treatment, and the crude fat content increased by 5.61% from low to medium irrigation levels, but decreased by 1.95% from medium to sufficient irrigation levels.

### 3.6. Construction of a Comprehensive Growth Evaluation System for Maize in the Strip Intercropping System

#### 3.6.1. Comprehensive Evaluation Hierarchical Model

A hierarchical model for the comprehensive evaluation of maize was established using Yaaph V10.3 software (Appendix A). The AHP method and the entropy weight method were employed to assign weights to individual indicators of maize, and the weights of each maize indicator were calculated (Appendix A). In order to improve the reliability and scientific accuracy of the weight assignment values, and to avoid the influence of subjective factors on the evaluation, a basic weight set was constructed on the basis of the two assignment values obtained (Appendix A). The evaluation based on the TOPSIS comprehensive model with combined weighting involves normalizing the decision matrix, constructing a weighted matrix, and then calculating the ideal solution and closeness coefficient for each evaluation indicator (Table 7). From Table 7, it can be observed that the N2W3 (medium nitrogen and sufficient water levels) treatment achieved the highest closeness coefficient for maize, indicating the best comprehensive performance, followed by the N3W2 (high nitrogen and medium water levels) and N3W3 (high nitrogen and sufficient water levels) treatments. In contrast, the N1W1 (low nitrogen and low water levels) treatment had the lowest closeness coefficient, indicating that the comprehensive performance was the worst.

#### 3.6.2. Water–Nitrogen Coupling Response Model for Comprehensive Maize Growth

The comprehensive growth score (*y*) of maize was modeled through binary quadratic regression simulation with the coded irrigation amount (*x*_1_) and the coded nitrogen application rate (*x*_2_) as predictors, resulting in the following equation:$$y=0.139+1.021x_{1}+0.534x_{2}-0.614x_{1}^{2}-0.395x_{2}^{2}+0.051x_{1}x_{2}$$

A significance test was conducted on the regression equation. The coefficient of determination *R*^2^ = 0.977, indicating an excellent goodness of fit. The F-statistic (*F* = 12.736, *p* < 0.05) demonstrated that the regression relationship reached statistical significance, demonstrating the reliability of the established regression model.

The integrated growth of maize was influenced by the coupling effect of the irrigation amount and the nitrogen application rate. Based on the established regression equation, a three-dimensional interaction plot (Figure 5) was created to illustrate the combined effects of irrigation and nitrogen application on the comprehensive growth indicators of maize. According to the regression equation, the maximum comprehensive score (*y*) was 0.7743 when *x*_1_ was 0.8618 and *x*_2_ was 0.7316. This corresponds to an irrigation amount of 562.11 mm in 2023 and 549.15 mm in 2024, with a nitrogen application rate of 389.61 kg ha^−1^. As shown in Figure 5, the water–nitrogen coupling closure region was delineated based on 90% of the maximum comprehensive score. This closed region occurred at medium–sufficient irrigation levels and medium nitrogen application levels, indicating that the optimal irrigation amount for agricultural production was 525.15~578.32 mm (2023) and 513.06~565.00 mm (2024), while the optimal nitrogen application rate was 354.78~422.51 kg ha^−1^.

### 3.7. Construction of a Comprehensive Growth Evaluation System for Soybean in the Strip Intercropping System

#### 3.7.1. Comprehensive Evaluation Hierarchical Model

A hierarchical model for the comprehensive evaluation of soybean was established using Yaaph V10.3 software (Appendix A). The AHP method and the entropy weight method were employed to assign weights to the individual indicators of soybeans, and the weights of each soybean indicator were calculated (Appendix A). In order to improve the reliability and scientific accuracy of the weight assignment values and avoid the influence of subjective factors on the evaluation, a basic weight set was constructed on the basis of the two assignment values obtained (Appendix A). The evaluation based on the TOPSIS comprehensive model with combined weighting involves normalizing the decision matrix, constructing a weighted matrix, and then calculating the ideal solution and closeness coefficient for each evaluation indicator (Table 8). From Table 8, it can be observed that the N2W3 (medium nitrogen and sufficient water levels) treatment achieved the highest closeness coefficient for soybean, indicating the best comprehensive performance, followed by the N3W3 (high nitrogen and sufficient water levels) and N3W2 (high nitrogen and medium water levels) treatments. In contrast, the N1W1 (low nitrogen and low water levels) treatment had the lowest closeness coefficient, indicating that the comprehensive performance was the worst.

#### 3.7.2. Water–Nitrogen Coupling Response Model for Comprehensive Soybean Growth

The comprehensive growth score (*y*) of soybean was modeled through binary quadratic regression simulation with the coded irrigation amount (*x*_1_) and the coded nitrogen application rate (*x*_2_) as predictors, resulting in the following equation:$$y=0.095+1.044x_{1}+0.741x_{2}-0.576x_{1}^{2}-0.496x_{2}^{2}-0.014x_{1}x_{2}$$

A significance test was conducted on the regression equation. The coefficient of determination *R*^2^ = 0.971, indicating an excellent goodness of fit. The F-statistic (*F* =9.762, *p* < 0.05) demonstrated that the regression relationship reached statistical significance, demonstrating the reliability of the established regression model.

The integrated growth of soybean was influenced by the coupling effect of the irrigation amount and the nitrogen application rate. Based on the established regression equation, a three-dimensional interaction plot (Figure 6) was created to illustrate the combined effects of irrigation and nitrogen application on the comprehensive growth indicators of soybean. According to the regression equation, the maximum comprehensive score (*y*) was 0.5686 when *x*_1_ was 0.9063 and *x*_2_ was 0.0007. This corresponds to an irrigation amount of 81.35 mm in 2023 and 90.17 mm in 2024, with a nitrogen application rate of 42.53 kg ha^−1^. As shown in Figure 6, the water–nitrogen coupling closure region was delineated based on 90% of the maximum comprehensive score. This closed region occurred at medium–sufficient irrigation levels and medium nitrogen application levels, indicating that the optimal irrigation amount for agricultural production was 69.13~76.47 mm (2023) and 76.17~84.25 mm (2024), while the optimal nitrogen application rate was 60.27~71.81 kg ha^−1^.

## 4. Discussion

### 4.1. Water–Nitrogen Interaction Effects on Yield Components and Yields of Maize and Soybean in the Strip Intercropping System

Water and nitrogen are essential elements for crop growth and development, and the interaction between the two factors is closely related to crop growth. An appropriate water–nitrogen ratio can effectively enhance fertilizer efficiency, promote crop growth, improve yield components, and increase dry matter accumulation and yield [25,26,27]. Siva Lakshmi et al. [28] found that appropriate water and nitrogen rates could improve maize yield components and increase 100-kernel weight, but the yield increasing effect gradually weakened with a higher irrigation amount and nitrogen application rate. Ma et al. [29] demonstrated that excessive irrigation and nitrogen application both reduced maize 100-kernel weight and kernel number per ear. Therefore, appropriately reducing water and nitrogen inputs can effectively increase the yield components. The results of this study have demonstrated that an optimal water–nitrogen combination could effectively improve the hydro-fertilizer environment for maize growth, enhance dry matter accumulation, and optimize yield components, thereby laying the foundation for high maize yields. These findings further validate the research conclusions of Siva Lakshmi et al. [28] and Ma et al. [29]. Furthermore, the results of this study have revealed that the medium nitrogen and sufficient water level (N2W3) treatment achieved the highest maize yields, reaching 14,027.21 kg ha^−1^ in 2023 and 13,986.83 kg ha^−1^ in 2024, respectivley, representing an increase of 2.52%~138.85% compared to that of the other treatments. This indicates a coupling interaction between water and nitrogen in crop growth, demonstrating that maximum yield is not necessarily achieved by simply sufficient irrigation and nitrogen application. These findings align with the research of Saed-Moucheshi et al. [30] and Sinclair et al. [31], though the magnitude of yield improvement varied slightly due to differences in cropping systems, water–nitrogen management practices, and the environment of the experimental areas. Li et al. [32] confirmed through research in a maize–wheat strip intercropping system in a semi-arid area that supplementary irrigation and optimal nitrogen application alleviated the interspecific competition, promoted the recovery growth of intercropped maize, and improved the yield of the wheat/maize intercropping system. Moreover, the nitrogen application rate of maize was 300 kg ha^−1^, which was similar to that of the nitrogen fertilizer dosage in this experiment, but the supplementary irrigation amount was significantly lower than that used in this experiment. Therefore, in semi-arid and sub-humid areas, where water is a limiting factor and the effect of nitrogen fertilizer is limited by water, we must strengthen the management of water, and water management should take precedence over nitrogen fertilizer.

Bai et al. [33] found that, under single-factor conditions (either water or nitrogen), simultaneously increasing the irrigation amount and nitrogen application rate significantly improved soybean yield and 100-kernel weight but had limited effects on the pod number per plant or the kernel number per plant. However, under water–nitrogen interaction, the low nitrogen and sufficient water level (N1W3) combination demonstrated superior coupling effects, resulting in the highest soybean yield. Lu et al. [34] demonstrated that the effects of water and nitrogen treatments on soybean yield are not isolated, but rather interact and influence each other. The highest soybean yield was achieved under the medium nitrogen and sufficient water level (N2W3) treatment, not the high nitrogen and sufficient water level (N3W3) treatment. This is due to excessive nitrogen application, which leads to overdeveloped vegetative growth in soybean plants, negatively affecting pod formation and ultimately resulting in sub-optimal kernel yield. This study has revealed that both the irrigation amount and the nitrogen application rate significantly influenced the soybean yield components and final yield. Appropriate water–nitrogen interaction could effectively optimize the yield components, thereby enhancing both yield and dry matter accumulation. When the irrigation amount was 100% ET_0_, as the nitrogen application rate increased, the number of effective pods per plant, the kernel weight per plant, the kernel number per plant, the 100-kernel weight, the yield, and the dry matter accumulation of soybeans all showed a trend of initially increasing and subsequently decreasing. Notably, the nitrogen effect surpassed the irrigation effect in magnitude. The highest yield and dry matter accumulation were achieved at a nitrogen application rate of 57 kg ha^−1^, with two-year averages of 2025.39 kg ha^−1^ and 5431.84 kg ha^−1^, respectively, which were 5.37%~191.44% and 4.72%~319.23% higher than those of the no nitrogen, low nitrogen, and high nitrogen treatments, suggesting that blindly and continuously increasing nitrogen application rates does not lead to higher yields. Instead, this might have caused excessive vegetative growth, resulting in lodging, reduced pod formation, and other growth imbalances, ultimately lowering the yield, which also corroborated the findings of researchers such as Bai et al. [33] and Lu et al. [34].

### 4.2. Water–Nitrogen Interaction Effects on Water–Nitrogen Productivity of Maize and Soybean in the Strip Intercropping System

The unreasonable use of water and nitrogen not only leads to the waste of water resources and fertilizers, as well as a reduction in efficiency, but also causes ecological problems such as soil compaction and salinization [35,36,37]. Particularly in the Hexi Corrido, where saline–alkali land is widely distributed and the ecological environment is fragile, it is particularly important to reasonably control the amount of fertilizer application. Xing et al. [38] found that reducing the irrigation amount while increasing the nitrogen application rate can improve crop water productivity, whereas the opposite approach enhances the crop partial factor productivity of the fertilizer. In this study, increasing the irrigation amount improved the NPF and NAE in the strip intercropping of maize and soybeans. However, excessive water input reduced both WP and IP, which aligns with the findings of Eissa et al. [39], Ran et al. [40], and Liao et al. [11], further corroborating the conclusions of Xing et al. [38]. Under the same irrigation level, different nitrogen application rates led to varying performances in WP and IP. Specifically, under low water conditions, as the nitrogen application increased, the WP of maize and soybean showed a trend of initially increasing and subsequently decreasing, indicating a threshold effect of the nitrogen application rate on WP enhancement. Within this threshold, WP showed a positive correlation with the nitrogen application rate but a negative correlation beyond it. In contrast, IP exhibited a continuous increasing trend. Under medium irrigation conditions, both the WP and IP of maize and soybean increased steadily with higher nitrogen application rates. Under sufficient irrigation conditions, IP showed a trend of initially increasing and subsequently decreasing with a rising nitrogen application rate, suggesting a threshold effect of the nitrogen application rate on IP enhancement—positive below the threshold and negative above it. In contrast, WP showed a consistent increasing trend throughout the nitrogen application gradient. These findings align with the research results of Momen et al. [41] and Lu et al. [34]. In this experiment, under water–nitrogen interaction, higher water–nitrogen productivity did not demonstrate consistency. Specifically, the higher WP and IP of both maize and soybean were observed under the high nitrogen and medium water level (N3W2) treatment, while the higher NPF occurred under the low nitrogen and sufficient water level (N1W3) treatment. The NAE value was achieved under the medium nitrogen and sufficient water level (N2W3) treatment. Therefore, subsequent research should comprehensively balance water and nitrogen application rates in order to further clarify the optimal adaptation range between irrigation amount and nitrogen application rate. This approach aims to achieve synergistic effects of “water-enhanced fertilization and fertilizer-regulated irrigation,” ultimately realizing the coordinated improvement of both crop WP and NUE. Maize–soybean intercropping systems are widespread in the semi-arid to semi-humid climate zone of China. Supplemental irrigation of 30 mm can enhance the nitrogen complementarity effect, productivity, and resource use efficiency, while a nitrogen application rate of 180 kg ha^−1^ for maize can improve the NPF without reducing crop yield or WP [42]. The irrigation amount and nitrogen application rate are significantly lower than those in the experiment of this study. The reason for this is that the semi-arid area receives an average annual precipitation of 595 mm, compared to only 200 mm in the Hexi Oasis irrigation area, and the planting densities of maize and soybean are significantly lower than those in this experiment. Therefore, different planting areas should adopt appropriate water and nitrogen management strategies based on the local climatic conditions and planting densities. This approach can achieve efficient resource utilization while simultaneously improving both yield and quality.

### 4.3. Water–Nitrogen Interaction Effects on the Quality of Maize and Soybean in the Strip Intercropping System

Irrigation is one of the most frequent operations in farmland management in the oasis agricultural areas of the Hexi Corridor, directly affecting crop growth, yield, and quality formation [43]. Th existing research has confirmed that water management has a certain impact on the later growth stages of crops, particularly during the reproductive growth stage, where either excessive or insufficient irrigation can significantly affect kernel quality [44]. Nitrogen plays a vital role in crop metabolism, serving as a key component of various metabolically active substances, as well as a major structural element of cells, proteins, and enzymes [45]. Therefore, nitrogen also significantly influences crop growth, yield, and quality [46]. The existing research has shown that excessive nitrogen application leads to overly vigorous vegetative growth, increased lodging susceptibility, and reduced stress resistance in plants, ultimately impairing reproductive growth and consequently reducing crop yield and quality. Conversely, insufficient nitrogen application results in stunted plant growth, weakened resistance, decreased chlorophyll synthesis in the leaves, diminished photosynthetic activity, and negatively affects the accumulation of nutritional components in crops [47]. A study by Qu et al. [48] showed that, under the same irrigation level, the total starch and crude protein content in spring maize kernels showed an initially increasing and subsequently decreasing trend with rising nitrogen application rates, while the crude fat content exhibited a decreasing trend. Under the same nitrogen application conditions, both the crude protein and crude fat content decreased with rising irrigation, whereas the total starch content showed an increasing trend. The study by Wang et al. [49] demonstrated that, under medium nitrogen and medium water (N2W2) conditions, maize could maintain relatively high contents of crude protein, starch, and crude ash, while showing no significant effect on the crude fat content. The results of this study have indicated that, under different water and nitrogen conditions, the responses of the maize kernel quality indicators to water and nitrogen supply varied. Under the same irrigation level, as the nitrogen application rate increased, the crude fat, starch, crude protein, and lysine content in maize kernels showed a trend of initially increasing and subsequently decreasing, while the bulk density showed a continuous upward trend. Under the same nitrogen application level, the different quality indicators of maize kernels also exhibited varying responses to the irrigation amount. Under the no nitrogen level, maize kernel quality showed an increasing trend as the irrigation amount increased. Under low and medium nitrogen levels, the crude fat content showed a trend of initially increasing and subsequently decreasing with a higher irrigation amount; moreover, the content of starch and lysine and bulk density showed an increasing trend with a higher irrigation amount, while the crude protein content decreased as irrigation increased. Under sufficient nitrogen levels, the crude fat content initially increased and subsequently decreased with a higher irrigation amount; the starch content and bulk density exhibited an increasing trend with a higher irrigation amount; and the crude protein and lysine content initially increased and subsequently decreased as irrigation increased. In this experiment, some results were consistent with the findings of Qu et al. [48], Wang et al. [49], and Jahangirlou et al. [50], but there were still certain discrepancies. On the one hand, it may be that under varying environmental conditions in different experimental zones, differences in soil accumulated temperature may limit the kernel filling rate of maize, resulting in changes in the proportions of starch and crude protein content in the kernels. Simultaneously, region-specific soil variations directly affect maize root development, nutrient absorption, and secretion components, thereby influencing the accumulation of crude fat and lysine in the kernels. On the other hand, in this experiment, the maize–soybean strip intercropping system created competition and the redistribution of light resources among the crops, making the light distribution within the maize canopy more uniform, increasing the net photosynthetic rate, and enhancing carbon assimilation. Additionally, the nitrogen fixation of soybeans was released into the soil through root exudates, replenishing the soil nitrogen pool and optimizing the microecological environment. Meanwhile, different water and nitrogen usages affected maize carbon and nitrogen metabolism, as well as kernel filling, ultimately leading to variations in kernel starch, crude protein, and crude fat content.

The study by Pei et al. [51] showed that reducing the irrigation amount and increasing the nitrogen application rate can effectively improve soybean protein content, while the change in the fat content is opposite to that of the protein content. Wang et al. [52] confirmed that increasing the irrigation amount is unfavorable for the improvement of soybean protein and fat content, whereas increasing the nitrogen application rate can promote soybean protein content but is not conducive to fat accumulation. The results of this study indicated that, under the same irrigation level, increasing the nitrogen application rate can promote an increase in soybean crude protein content, while the crude fat content shows a trend of initially increasing and subsequently decreasing. Under the same nitrogen application level, the soybean quality indicators also exhibited differential responses to the irrigation amount. Under the no nitrogen condition, the soybean quality showed an increasing trend with a higher irrigation amount. Under low and high nitrogen conditions, the crude protein content showed a trend of initially decreasing and subsequently increasing with a higher irrigation amount, while the crude fat content showed a trend of initially increasing and subsequently decreasing. Under medium nitrogen conditions, the crude protein content showed a trend of initially decreasing and subsequently increasing with a higher irrigation amount, whereas the crude fat content demonstrated a consistently increasing trend. The results of this experiment are consistent with the findings of Panasiewicz et al. [53] and Kresović et al. [54] and also corroborate the research conclusions of Pei et al. [51] and Wang et al. [52]. Under the interaction of water and nitrogen, the highest crude protein content in soybeans was achieved with the high nitrogen and low water level (N3W1) treatment, while the highest crude fat content was obtained with the medium nitrogen and sufficient water level (N2W3) treatment. This aligns with Wang et al.’s [52] findings regarding crude protein content but shows slight differences in crude fat content, which may be attributed to factors such as variety, environmental conditions, and planting systems.

## 5. Conclusions

In the oasis agricultural area of the Hexi Corridor, different water–nitrogen combinations demonstrated significant interactive effects on yield and yield components, water–nitrogen productivity, NAE, and kernel quality in the maize–soybean strip intercropping system. Considering the growth environment of the oasis agricultural area in the Hexi Corridor and the comprehensive effects of the water–nitrogen supply on yield, water–nitrogen productivity, and kernel quality in maize–soybean strip intercropping systems, the optimal management strategy was determined to be as follows: when the irrigation quota of maize and soybean is 100% ET_0_, with nitrogen application rates of 354.78~422.51 kg ha^−1^ and 60.27~71.81 kg ha^−1^, respectively, the highest target yields can be achieved. This optimal irrigation and fertilization regime not only ensures high water–nitrogen productivity, but also produces superior-quality kernels. However, the conditions of sufficient water level and high nitrogen application disrupted the carbon–nitrogen metabolic balance of crops, and the carbon resources were utilized by non-essential amino acids, inhibiting the lysine synthesis pathway in maize kernels. Meanwhile, nitrogen surplus caused soybean photosynthetic products to preferentially convert into proteins, ultimately leading to a decline in crude fat content. These findings provide practical guidance for maize–soybean strip intercropping systems in the Hexi Corridor oasis agricultural area, effectively balancing the dual objectives of high yield and kernel quality. However, this study was limited to investigating the dual-factor regulation of irrigation amount and nitrogen application rate, without systematically examining the synergistic effects between the irrigation amount and the NPK (nitrogen–phosphorus–potassium) application ratios. Follow-up research will focus on precisely quantifying these water–fertilizer coupling relationships to establish more scientific and rational irrigation and fertilization regimes for maize–soybean strip intercropping systems in the Hexi Corridor oasis agricultural area.

## Figures and Tables

**Figure 1 plants-14-02050-f001:**
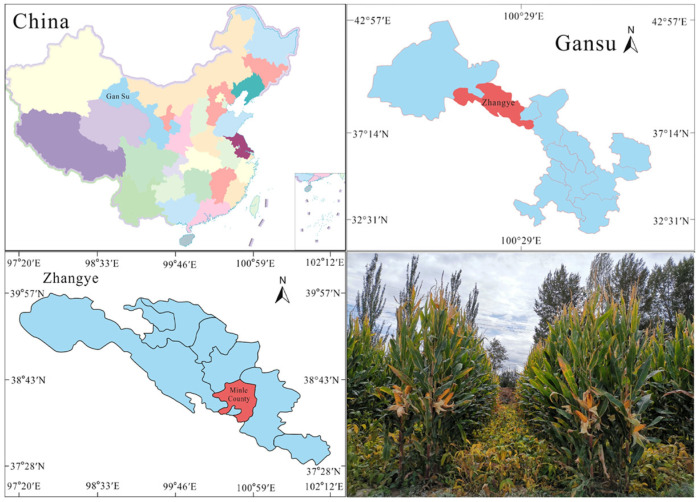
Location of the experimental site.

**Figure 2 plants-14-02050-f002:**
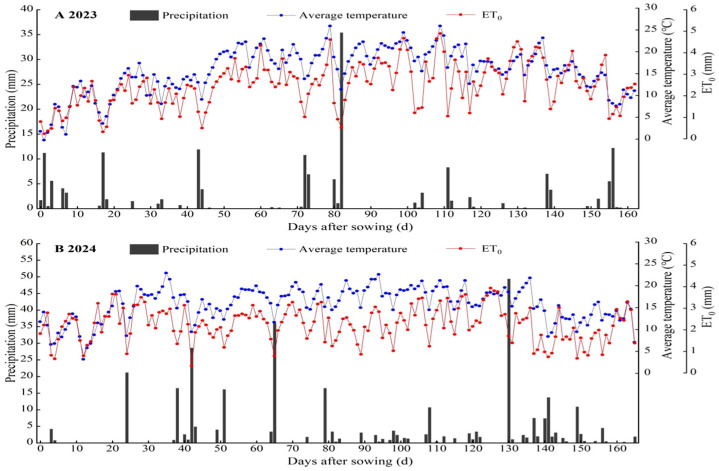
Daily variation of average temperature, reference crop evapotranspiration (ET_0_), and precipitation throughout the maize and soybean growing seasons in the strip intercropping system of 2023 (**A**) and 2024 (**B**).

**Figure 3 plants-14-02050-f003:**
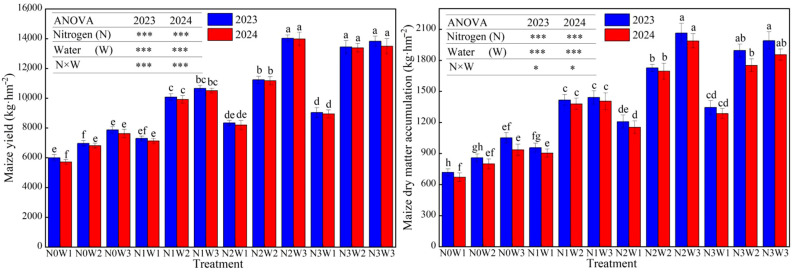
Water–nitrogen interaction effects on maize yield and dry matter accumulation in the strip intercropping system. Bars and error bars stand to represent averaged values ± standard errors (*n* = 3), different lowercase letters indicate significant differences among the treatments (*p* < 0.05), the * and *** indicate significant differences among the different treatments at the levels of *p* < 0.05 and *p* < 0.001.

**Figure 4 plants-14-02050-f004:**
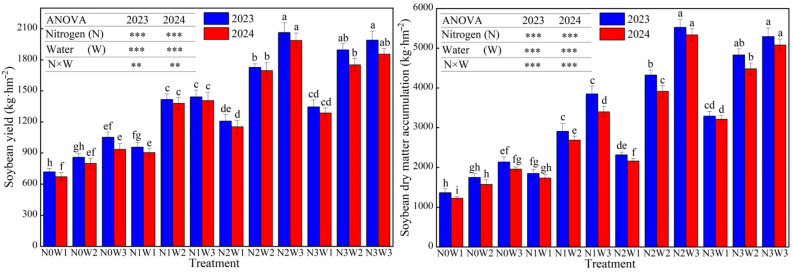
Water–nitrogen interaction effects on soybean yield and dry matter accumulation in the strip intercropping system. Bars and error bars stand to represent averaged values ± standard errors (n = 3), different lowercase letters indicate significant differences among the treatments (*p* < 0.05), the ** and *** indicate significant differences among the different treatments at the levels of *p* < 0.01 and *p* < 0.001.

**Figure 5 plants-14-02050-f005:**
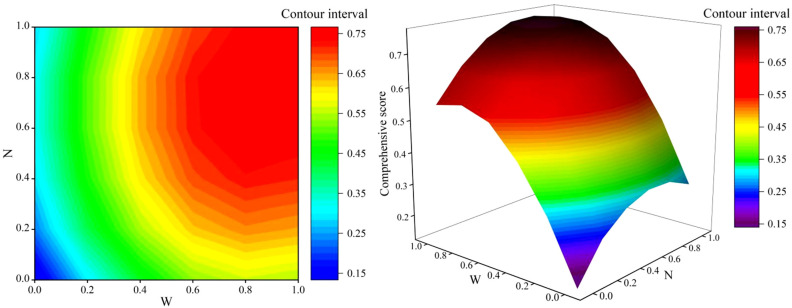
The effect of water–nitrogen coupling on the comprehensive growth of maize in the strip intercropping system.

**Figure 6 plants-14-02050-f006:**
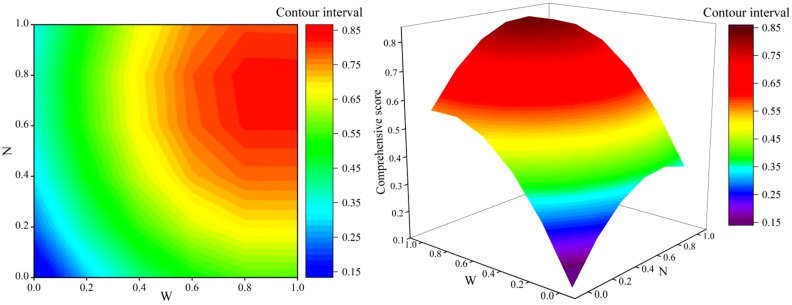
The effect of water–nitrogen coupling on the comprehensive growth of soybean in the strip intercropping system.

**Table 1 plants-14-02050-t001:** Experimental design and factor coding values.

Treatment	Irrigation Quota (mm)				Total Nitrogen Application (kg ha^−1^)		Factor Coding Values	
	Maize		Soybean		Maize	Soybean	Irrigation Amount x_1_	Nitrogen Application Rate x_2_
	2023 Year	2024 Year	2023 Year	2024 Year				
N0W1	357.0	348.8	50.7	55.86	0	0	—	—
N0W2	476.0	465.0	67.6	74.48	0	0	—	—
N0W3	595.0	581.3	84.5	93.10	0	0	—	—
N1W1	357.0	348.8	50.7	55.86	225.0	42.5	0.0	0.0
N1W2	476.0	465.0	67.6	74.48	225.0	42.5	0.5	0.0
N1W3	595.0	581.3	84.5	93.10	225.0	42.5	1.0	0.0
N2W1	357.0	348.8	50.7	55.86	337.5	63.8	0.0	0.5
N2W2	476.0	465.0	67.6	74.48	337.5	63.8	0.5	0.5
N2W3	595.0	581.3	84.5	93.10	337.5	63.8	1.0	0.5
N3W1	357.0	348.8	50.7	55.86	450.0	85.0	0.0	1.0
N3W2	476.0	465.0	67.6	74.48	450.0	85.0	0.5	1.0
N3W3	595.0	581.3	84.5	93.10	450.0	85.0	1.0	1.0

**Table 2 plants-14-02050-t002:** Water–nitrogen interaction effects on maize yield components in the strip intercropping system.

Year	Treatment	Ear Length (cm)	Ear Diameter (mm)	Kernel Number per Ear (a)	Ear Weight (g)	100-Kernel Weight (g)
2023	N0W1	17.05 ± 0.38 ^h^	43.74 ± 0.93 ^f^	504 ± 17.98 ^g^	149.76 ± 2.29 ^g^	27.78 ± 0.68 ^e^
	N0W2	17.80 ± 0.51 ^gh^	45.05 ± 1.09 ^ef^	522 ± 11.09 ^g^	167.77 ± 4.99 ^fg^	30.96 ± 1.00 ^de^
	N0W3	18.62 ± 0.38 ^efg^	46.82 ± 0.23 ^de^	576 ± 10.26 ^ef^	174.63 ± 2.56 ^ef^	31.48 ± 1.19 ^d^
	N1W1	18.24 ± 0.34 ^fgh^	46.30 ± 0.53 ^de^	544 ± 9.21 ^fg^	161.27 ± 3.67 ^fg^	29.88 ± 0.63 ^de^
	N1W2	20.16 ± 0.41 ^cde^	48.97 ± 0.58 ^bc^	624 ± 16.59 ^de^	197.15 ± 8.63 ^cd^	35.06 ± 1.25 ^bc^
	N1W3	20.78 ± 0.60 ^bcd^	49.61 ± 0.63 ^abc^	672 ± 13.12 ^cd^	215.16 ± 3.57 ^c^	35.40 ± 1.41 ^bc^
	N2W1	19.09 ± 0.56 ^efg^	47.48 ± 0.31 ^cd^	594 ± 20.01 ^e^	186.18 ± 6.34 ^de^	32.17 ± 0.51 ^cd^
	N2W2	21.27 ± 0.46 ^abc^	50.25 ± 0.48 ^ab^	702 ± 13.03 ^bc^	239.51 ± 6.08 ^b^	36.95 ± 1.36 ^ab^
	N2W3	22.83 ± 0.62 ^a^	51.74 ± 0.79 ^a^	756 ± 22.49 ^a^	262.01 ± 8.45 ^a^	38.92 ± 1.10 ^a^
	N3W1	19.71 ± 0.35 ^def^	48.01 ± 0.59 ^cd^	648 ± 14.34 ^d^	208.89 ± 6.21 ^c^	32.71 ± 1.06 ^cd^
	N3W2	21.85 ± 0.48 ^ab^	50.83 ± 0.58 ^ab^	720 ± 20.00 ^abc^	248.53 ± 6.59 ^ab^	37.94 ± 1.13 ^ab^
	N3W3	22.41 ± 0.68 ^a^	51.36 ± 0.84 ^a^	738 ± 13.77 ^ab^	254.48 ± 7.18 ^ab^	38.06 ± 0.66 ^ab^
	W	***	***	***	***	***
	N	***	***	***	***	***
	W × N	*	ns	ns	*	ns
2024	N0W1	16.56 ± 0.23 ^h^	43.01 ± 1.17 ^e^	486 ± 12.09 ^i^	138.54 ± 4.63 ^h^	25.21 ± 0.78 ^g^
	N0W2	17.08 ± 0.38 ^gh^	44.40 ± 1.30 ^de^	510 ± 14.37 ^i^	159.07 ± 5.16 ^fg^	28.18 ± 0.61 ^f^
	N0W3	17.90 ± 0.29 ^fg^	45.72 ± 0.41 ^cde^	560 ± 12.08 ^gh^	167.09 ± 6.03 ^efg^	29.40 ± 1.04 ^ef^
	N1W1	17.61 ± 0.49 ^fh^	45.19 ± 1.12 ^cde^	532 ± 18.51 ^hi^	150.76 ± 3.29 ^gh^	27.16 ± 0.89 ^fg^
	N1W2	19.73 ± 0.55 ^cd^	48.09 ± 0.66 ^abc^	612 ± 17.25 ^ef^	182.74 ± 5.35 ^de^	33.51 ± 0.75 ^cd^
	N1W3	20.15 ± 0.37 ^cd^	48.85 ± 0.61 ^ab^	666 ± 17.97 ^cd^	205.15 ± 5.52 ^c^	34.11 ± 0.59 ^bc^
	N2W1	18.42 ± 0.40 ^ef^	46.74 ± 0.42 ^bcd^	588 ± 18.90 ^fg^	171.89 ± 6.44 ^ef^	30.89 ± 1.02 ^e^
	N2W2	20.84 ± 0.33 ^bc^	49.06 ± 0.58 ^ab^	688 ± 13.89 ^c^	227.66 ± 6.82 ^b^	36.09 ± 0.88 ^ab^
	N2W3	22.57 ± 0.41 ^a^	50.93 ± 0.98 ^a^	752 ± 13.60 ^a^	257.40 ± 8.88 ^a^	37.53 ± 0.51 ^a^
	N3W1	19.14 ± 0.30 ^de^	47.28 ± 0.46 ^bcd^	640 ± 16.34 ^de^	192.13 ± 6.59 ^cd^	31.25 ± 0.79 ^de^
	N3W2	21.39 ± 0.22 ^ab^	49.71 ± 1.26 ^ab^	704 ± 15.02 ^bc^	239.82 ± 7.53 ^ab^	36.54 ± 0.88 ^ab^
	N3W3	22.03 ± 0.52 ^a^	50.37 ± 1.31 ^a^	736 ± 14.14 ^ab^	251.85 ± 8.23 ^a^	37.28 ± 1.07 ^a^
	W	***	***	***	***	***
	N	***	***	***	**	***
	W × N	*	ns	ns	*	ns
ANOVA						
Year (Y)		*	*	ns	***	***
Water (W)		***	***	***	***	***
Nitrogen (N)		***	*	***	***	**
Y × W		ns	ns	ns	ns	ns
Y × N		ns	ns	ns	ns	ns
W × N		***	***	***	***	***
Y × W × N		ns	ns	ns	ns	ns

Note: Different lowercase letters in the same column indicate significant differences among the different treatments (*p* < 0.05). The *, **, and *** indicate significant differences among the different treatments at the levels of *p* < 0.05, *p* < 0.01, and *p* < 0.001, respectively. The ns means not significant at the level of *p* ≥ 0.05.

**Table 3 plants-14-02050-t003:** Water–nitrogen interaction effects on soybean yield components in the strip intercropping system.

Year	Treatment	Effective Pod Number per Plant (a)	Kernel Weight per Plant (g)	Kernel Number per Plant (a)	100-Kernel Weight (g)
2023	N0W1	21.98 ± 0.79 ^f^	7.15 ± 0.23 ^i^	50.90 ± 1.87 ^h^	9.42 ± 0.26 ^i^
	N0W2	24.15 ± 0.61 ^e^	7.76 ± 0.37 ^hi^	57.21 ± 2.16 ^gh^	10.05 ± 0.31 ^hi^
	N0W3	27.83 ± 0.53 ^d^	9.32 ± 0.46 ^gh^	65.64 ± 2.13 ^ef^	12.83 ± 0.44 ^fg^
	N1W1	25.79 ± 0.67 ^e^	8.59 ± 0.41 ^hi^	62.25 ± 2.10 ^fg^	11.46 ± 0.60 ^gh^
	N1W2	31.84 ± 0.50 ^c^	11.85 ± 0.47 ^ef^	75.99 ± 2.69 ^cd^	14.82 ± 0.54 ^de^
	N1W3	32.73 ± 0.66 ^bc^	13.69 ± 0.45 ^cd^	79.04 ± 2.00 ^bc^	15.91 ± 0.52 ^cd^
	N2W1	29.15 ± 0.64 ^d^	10.37 ± 0.52 ^fg^	70.74 ± 2.05 ^de^	13.17 ± 0.48 ^f^
	N2W2	34.01 ± 0.57 ^ab^	14.73 ± 0.51 ^bc^	83.24 ± 1.83 ^ab^	17.28 ± 0.70 ^bc^
	N2W3	36.08 ± 0.65 ^a^	17.24 ± 0.77 ^a^	89.50 ± 2.79 ^a^	19.05 ± 0.60 ^a^
	N3W1	29.81 ± 0.88 ^d^	13.04 ± 0.50 ^de^	73.01 ± 2.12 ^cd^	13.94 ± 0.46 ^ef^
	N3W2	34.72 ± 0.97 ^ab^	15.55 ± 0.59 ^b^	85.96 ± 3.04 ^ab^	18.09 ± 0.56 ^ab^
	N3W3	35.49 ± 0.59 ^a^	16.28 ± 0.87 ^ab^	87.43 ± 2.43 ^a^	18.62 ± 0.70 ^ab^
	W	***	***	***	***
	N	***	***	***	***
	W × N	ns	**	ns	*
2024	N0W1	19.87 ± 0.73 ^h^	6.16 ± 0.22 ^h^	44.81 ± 1.38 ^h^	8.65 ± 0.31 ^f^
	N0W2	22.31 ± 0.71 ^g^	6.80 ± 0.20 ^h^	52.72 ± 1.47 ^g^	9.49 ± 0.45 ^f^
	N0W3	24.90 ± 0.52 ^f^	8.63 ± 0.24 ^fg^	62.28 ± 1.29 ^f^	11.70 ± 0.41 ^de^
	N1W1	23.72 ± 0.83 ^fg^	8.02 ± 0.26 ^g^	60.5 ± 1.98 ^f^	10.21 ± 0.52 ^ef^
	N1W2	30.63 ± 0.92 ^cd^	10.56 ± 0.36 ^de^	75.06 ± 1.63 ^cd^	13.88 ± 0.51 ^bc^
	N1W3	31.84 ± 0.82 ^bc^	13.32 ± 0.46 ^c^	77.17 ± 2.64 ^cd^	15.07 ± 0.66 ^b^
	N2W1	27.26 ± 0.85 ^e^	9.71 ± 0.36 ^ef^	69.31 ± 0.95 ^e^	12.19 ± 0.57 ^cd^
	N2W2	33.28 ± 0.63 ^ab^	14.10 ± 0.39 ^bc^	80.42 ± 1.44 ^bc^	16.73 ± 0.70 ^a^
	N2W3	35.63 ± 1.06 ^a^	17.05 ± 0.58 ^a^	87.08 ± 2.66 ^a^	18.42 ± 0.53 ^a^
	N3W1	28.37 ± 0.77 ^de^	11.29 ± 0.36 ^d^	72.19 ± 1.68 ^de^	12.76 ± 0.39 ^cd^
	N3W2	34.05 ± 0.82 ^ab^	14.83 ± 0.50 ^b^	82.95 ± 1.94 ^ab^	17.18 ± 0.76 ^a^
	N3W3	34.92 ± 0.87 ^a^	16.47 ± 0.31 ^a^	84.73 ± 2.06 ^ab^	17.95 ± 0.62 ^a^
	W	***	***	***	***
	N	***	***	***	***
	W × N	ns	***	ns	*
ANOVA					
Year (Y)		***	***	**	***
Water (W)		***	***	***	***
Nitrogen (N)		***	***	***	***
Y × W		ns	ns	ns	ns
Y × N		ns	ns	ns	ns
W × N		***	***	***	***
Y × W × N		ns	ns	ns	ns

Note: Different lowercase letters in the same column indicate significant differences among the different treatments (*p* < 0.05). The *, **, and *** indicate significant differences among the different treatments at the levels of *p* < 0.05, *p* < 0.01, and *p* < 0.001, respectively. The ns means not significant at the level of *p* ≥ 0.05.

**Table 4 plants-14-02050-t004:** Water–nitrogen interaction effects on WP of maize and soybean in the strip intercropping system.

Year	Treatment	Maize			Soybean		
		ET (mm)	WP (kg m^−3^)	IP (kg m^−3^)	ET (mm)	WP (kg m^−3^)	IP (kg m^−3^)
2023	N0W1	499.55 ± 22.58 ^e^	1.20 ± 0.01 ^h^	1.68 ± 0.06 ^e^	145.45 ± 7.92 ^i^	0.50 ± 0.04 ^bcd^	1.42 ± 0.07 ^f^
	N0W2	546.33 ± 11.17 ^e^	1.28 ± 0.05 ^gh^	1.46 ± 0.04 ^f^	196.76 ± 9.70 ^fg^	0.44 ± 0.04 ^d^	1.27 ± 0.06 ^f^
	N0W3	658.14 ± 10.74 ^c^	1.20 ± 0.05 ^h^	1.32 ± 0.04 ^f^	259.09 ± 6.94 ^e^	0.41 ± 0.02 ^d^	1.24 ± 0.06 ^f^
	N1W1	536.29 ± 12.86 ^e^	1.36 ± 0.04 ^fg^	2.05 ± 0.04 ^d^	163.97 ± 8.40 ^hi^	0.59 ± 0.03 ^ab^	1.89 ± 0.08 ^de^
	N1W2	653.93 ± 12.90 ^c^	1.54 ± 0.04 ^cd^	2.12 ± 0.05 ^d^	245.68 ± 9.82 ^e^	0.58 ± 0.04 ^ab^	2.10 ± 0.08 ^d^
	N1W3	753.06 ± 19.93 ^b^	1.42 ± 0.02 ^ef^	1.79 ± 0.03 ^e^	316.43 ± 10.48 ^bc^	0.46 ± 0.04 ^cd^	1.71 ± 0.08 ^e^
	N2W1	542.15 ± 11.22 ^e^	1.54 ± 0.01 ^cd^	2.34 ± 0.05 ^c^	182.73 ± 11.14 ^gh^	0.67 ± 0.07 ^a^	2.38 ± 0.13 ^bc^
	N2W2	672.54 ± 17.00 ^c^	1.67 ± 0.04 ^b^	2.36 ± 0.05 ^c^	273.08 ± 11.42 ^de^	0.63 ± 0.03 ^a^	2.55 ± 0.05 ^abc^
	N2W3	821.98 ± 12.57 ^a^	1.71 ± 0.05 ^ab^	2.36 ± 0.04 ^c^	333.70 ± 9.41 ^ab^	0.62 ± 0.02 ^ab^	2.44 ± 0.11 ^bc^
	N3W1	605.95 ± 18.70 ^d^	1.49 ± 0.01 ^de^	2.53 ± 0.09 ^b^	216.05 ± 7.07 ^f^	0.62 ± 0.04 ^a^	2.65 ± 0.13 ^ab^
	N3W2	746.62 ± 19.78 ^b^	1.80 ± 0.05 ^a^	2.82 ± 0.09 ^a^	293.33 ± 12.70 ^cd^	0.65 ± 0.04 ^a^	2.80 ± 0.09 ^a^
	N3W3	837.50 ± 21.13 ^a^	1.65 ± 0.02 ^bc^	2.32 ± 0.06 ^c^	352.16 ± 10.43 ^a^	0.56 ± 0.01 ^abc^	2.35 ± 0.10 ^c^
	W	***	***	***	***	**	**
	N	***	***	***	***	***	***
	W × N	*	*	**	ns	ns	ns
2024	N0W1	478.00 ± 12.42 ^e^	1.20 ± 0.03 ^f^	1.64 ± 0.05 ^ef^	127.24 ± 9.16 ^h^	0.54 ± 0.07 ^abc^	1.20 ± 0.07 ^g^
	N0W2	521.39 ± 19.07 ^e^	1.31 ± 0.03 ^ef^	1.47 ± 0.04 ^fg^	188.07 ± 13.33 ^fg^	0.43 ± 0.02 ^cd^	1.07 ± 0.07 ^g^
	N0W3	630.17 ± 19.72 ^cd^	1.22 ± 0.08 ^f^	1.31 ± 0.05 ^g^	251.96 ± 12.43 ^cd^	0.38 ± 0.04 ^d^	1.01 ± 0.06 ^g^
	N1W1	502.76 ± 12.05 ^e^	1.42 ± 0.06 ^de^	2.04 ± 0.05 ^d^	153.54 ± 14.04 ^gh^	0.60 ± 0.06 ^ab^	1.62 ± 0.07 ^ef^
	N1W2	625.90 ± 12.10 ^cd^	1.59 ± 0.03 ^bc^	2.13 ± 0.05 ^d^	236.52 ± 10.74 ^de^	0.59 ± 0.04 ^ab^	1.85 ± 0.07 ^de^
	N1W3	737.40 ± 10.42 ^b^	1.43 ± 0.03 ^de^	1.81 ± 0.03 ^e^	309.71 ± 7.05 ^ab^	0.45 ± 0.02 ^bcd^	1.51 ± 0.09 ^f^
	N2W1	518.26 ± 21.00 ^e^	1.58 ± 0.07 ^bc^	2.35 ± 0.09 ^c^	175.80 ± 14.29 ^fg^	0.67 ± 0.08 ^a^	2.06 ± 0.11 ^bcd^
	N2W2	658.49 ± 22.38 ^c^	1.70 ± 0.06 ^ab^	2.40 ± 0.06 ^bc^	267.83 ± 11.46 ^cd^	0.64 ± 0.04 ^a^	2.28 ± 0.10 ^ab^
	N2W3	795.37 ± 25.92 ^a^	1.76 ± 0.02 ^a^	2.41 ± 0.08 ^bc^	324.55 ± 10.98 ^a^	0.61 ± 0.02 ^a^	2.13 ± 0.07 ^abc^
	N3W1	589.06 ± 19.35 ^d^	1.52 ± 0.02 ^cd^	2.57 ± 0.08 ^b^	211.63 ± 9.06 ^ef^	0.61 ± 0.04 ^a^	2.30 ± 0.09 ^ab^
	N3W2	729.71 ± 25.11 ^b^	1.84 ± 0.06 ^a^	2.88 ± 0.06 ^a^	276.60 ± 12.51 ^bc^	0.64 ± 0.05 ^a^	2.35 ± 0.08 ^a^
	N3W3	802.63 ± 20.52 ^a^	1.68 ± 0.06 ^ab^	2.32 ± 0.09 ^c^	340.70 ± 14.24 ^a^	0.55 ± 0.03 ^abc^	1.99 ± 0.06 ^cd^
	W	***	***	***	***	*	**
	N	***	***	***	***	***	***
	W × N	**	ns	**	ns	ns	ns
ANOVA							
Year (Y)		***	ns	ns	*	ns	***
Water (W)		***	***	***	***	***	***
Nitrogen (N)		***	***	***	**	**	***
Y × W		ns	ns	ns	ns	ns	ns
Y × N		ns	ns	ns	ns	ns	ns
W × N		***	***	***	***	***	***
Y × W × N		ns	ns	ns	ns	ns	ns

Note: Different lowercase letters in the same column indicate significant differences among the different treatments (*p* < 0.05). The *, **, and *** indicate significant differences among the different treatments at the levels of *p* < 0.05, *p* < 0.01, and *p* < 0.001, respectively. The ns means not significant at the level of *p* ≥ 0.05.

**Table 5 plants-14-02050-t005:** Water–nitrogen interaction effects on NUE of maize and soybean in the strip intercropping system.

Year	Treatment	Maize		Soybean	
		NPF (kg kg^−1^)	NAE (kg kg^−1^)	NPF (kg kg^−1^)	NAE (kg kg^−1^)
2023	N1W1	32.47 ± 0.66 ^de^	5.78 ± 0.62 ^c^	22.52 ± 0.99 ^c^	5.63 ± 0.41 ^e^
	N1W2	44.81 ± 0.96 ^b^	13.84 ± 1.59 ^b^	33.35 ± 1.24 ^a^	13.13 ± 0.90 ^b^
	N1W3	47.38 ± 0.88 ^a^	12.37 ± 0.67 ^b^	33.92 ± 1.49 ^a^	9.17 ± 0.82 ^d^
	N2W1	24.72 ± 0.48 ^g^	6.92 ± 0.49 ^c^	18.92 ± 1.00 ^d^	7.67 ± 0.84 ^d^
	N2W2	33.30 ± 0.70 ^d^	12.65 ± 0.91 ^b^	27.07 ± 0.52 ^b^	13.60 ± 0.51 ^b^
	N2W3	41.56 ± 0.69 ^c^	18.22 ± 0.63 ^a^	32.34 ± 1.48 ^a^	15.86 ± 0.68 ^a^
	N3W1	20.09 ± 0.74 ^h^	6.74 ± 1.22 ^c^	15.82 ± 0.77 ^d^	7.38 ± 0.39 ^de^
	N3W2	29.88 ± 0.95 ^f^	14.39 ± 0.74 ^b^	22.30 ± 0.71 ^c^	12.19 ± 0.26 ^bc^
	N3W3	30.73 ± 0.74 ^ef^	13.23 ± 1.14 ^b^	23.41 ± 1.01 ^c^	11.03 ± 0.42 ^c^
	W	***	***	***	***
	N	***	ns	***	***
	W × N	**	*	*	**
2024	N1W1	31.70 ± 0.74 ^cd^	6.26 ± 0.08 ^c^	21.28 ± 0.94 ^cd^	5.46 ± 0.20 ^e^
	N1W2	44.09 ± 1.14 ^ab^	13.81 ± 0.39 ^b^	32.46 ± 1.28 ^a^	13.63 ± 0.17 ^b^
	N1W3	46.70 ± 0.67 ^a^	12.78 ± 1.10 ^b^	33.07 ± 1.89 ^a^	11.04 ± 0.61 ^c^
	N2W1	24.31 ± 0.91 ^e^	7.35 ± 0.48 ^c^	18.08 ± 0.96 ^de^	7.54 ± 0.42 ^d^
	N2W2	33.13 ± 0.80 ^c^	12.94 ± 0.67 ^b^	26.57 ± 1.22 ^b^	14.02 ± 0.83 ^b^
	N2W3	41.44 ± 1.30 ^b^	18.83 ± 0.99 ^a^	31.15 ± 1.10 ^a^	16.48 ± 0.79 ^a^
	N3W1	19.90 ± 0.59 ^f^	7.18 ± 0.39 ^c^	15.13 ± 0.57 ^e^	7.22 ± 0.16 ^d^
	N3W2	29.75 ± 0.63 ^d^	14.61 ± 0.59 ^b^	20.61 ± 0.73 ^cd^	11.20 ± 0.20 ^c^
	N3W3	29.99 ± 1.14 ^d^	13.03 ± 0.52 ^b^	21.82 ± 0.64 ^c^	10.81 ± 0.06 ^c^
	W	***	***	***	***
	N	***	**	***	***
	W × N	**	***	*	***
ANOVA					
Year (Y)		ns	ns	*	ns
Water (W)		***	**	***	***
Nitrogen (N)		***	***	***	***
Y × W		ns	ns	ns	ns
Y × N		ns	ns	ns	ns
W × N		***	***	**	***
Y × W × N		ns	ns	ns	ns

Note: Different lowercase letters in the same column indicate significant differences among the different treatments (*p* < 0.05). The *, **, and *** indicate significant differences among the different treatments at the levels of *p* < 0.05, *p* < 0.01, and *p* < 0.001, respectively. The ns means not significant at the level of *p* ≥ 0.05.

**Table 6 plants-14-02050-t006:** Water–nitrogen interaction effects on the quality of maize and soybean in the strip intercropping system.

Year	Treatment	Maize					Soybean	
		Crude Fat (%)	Starch (%)	Crude Protein (%)	Lysine (%)	Bulk Density (g L^−1^)	Crude Protein (%)	Crude Fat (%)
2023	N0W1	3.47 ± 0.06 ^c^	44.90 ± 1.73 ^h^	6.55 ± 0.30 ^h^	0.21 ± 0.01 ^h^	673 ± 20.96 ^b^	38.04 ± 0.94 ^ab^	19.04 ± 0.98 ^b^
	N0W2	3.60 ± 0.13 ^bc^	49.81 ± 1.52 ^g^	6.97 ± 0.28 ^gh^	0.23 ± 0.01 ^gh^	684 ± 17.82 ^ab^	36.32 ± 1.25 ^ab^	19.87 ± 0.72 ^ab^
	N0W3	3.64 ± 0.04 ^bc^	54.05 ± 1.87 ^efg^	7.28 ± 0.34 ^fgh^	0.25 ± 0.01 ^fg^	701 ± 9.56 ^ab^	35.80 ± 1.35 ^b^	20.19 ± 0.39 ^ab^
	N1W1	3.84 ± 0.08 ^ab^	51.38 ± 1.82 ^fg^	9.87 ± 0.29 ^bc^	0.24 ± 0.01 ^g^	682 ± 9.69 ^ab^	38.51 ± 1.36 ^ab^	21.21 ± 0.79 ^ab^
	N1W2	4.02 ± 0.14 ^a^	61.96 ± 1.13 ^cd^	9.32 ± 0.30 ^cd^	0.29 ± 0.01 ^de^	694 ± 11.22 ^ab^	36.69 ± 1.31 ^ab^	21.85 ± 0.32 ^a^
	N1W3	3.73 ± 0.09 ^abc^	65.01 ± 2.19 ^bc^	8.75 ± 0.46 ^de^	0.30 ± 0.01 ^d^	711 ± 11.49 ^ab^	36.91 ± 1.53 ^ab^	20.58 ± 0.59 ^ab^
	N2W1	3.81 ± 0.08 ^abc^	58.47 ± 1.74 ^de^	11.05 ± 0.52 ^a^	0.36 ± 0.01 ^ab^	685 ± 18.20 ^ab^	39.13 ± 1.32 ^ab^	21.03 ± 0.37 ^ab^
	N2W2	3.95 ± 0.09 ^ab^	66.73 ± 0.96 ^abc^	10.73 ± 0.19 ^ab^	0.38 ± 0.02 ^a^	706 ± 10.65 ^ab^	37.05 ± 1.36 ^ab^	22.16 ± 0.90 ^a^
	N2W3	3.76 ± 0.10 ^abc^	71.29 ± 1.13 ^a^	8.42 ± 0.20 ^de^	0.33 ± 0.01 ^c^	721 ± 19.49 ^ab^	38.16 ± 1.08 ^ab^	22.37 ± 0.50 ^a^
	N3W1	3.95 ± 0.16 ^ab^	54.88 ± 1.43 ^ef^	8.06 ± 0.44 ^ef^	0.27 ± 0.01 ^ef^	692 ± 16.07 ^ab^	40.27 ± 1.64 ^a^	20.70 ± 0.87 ^ab^
	N3W2	3.90 ± 0.14 ^ab^	63.52 ± 1.09 ^c^	10.21 ± 0.20 ^abc^	0.35 ± 0.01 ^bc^	714 ± 9.58 ^ab^	37.84 ± 1.13 ^ab^	22.25 ± 0.89 ^a^
	N3W3	4.05 ± 0.08 ^a^	69.14 ± 1.64 ^ab^	7.69 ± 0.36 ^efg^	0.34 ± 0.01 ^bc^	725 ± 19.77 ^a^	38.72 ± 0.68 ^ab^	21.54 ± 1.03 ^a^
	W	0.99	65.56 ***	14.68 ***	24.04 ***	4.31 *	2.79 ns	2.05 ns
	N	7.89 **	58.63 ***	46.95 ***	102.05 ***	1.45 ns	1.71 ns	4.97 *
	W × N	1.06 ns	0.81 ns	7.84 ***	7.26 ***	0.04 ns	0.09 ns	0.47 ns
2024	N0W1	3.44 ± 0.21 ^b^	45.48 ± 1.01 ^h^	6.15 ± 0.50 ^g^	0.17 ± 0.01 ^i^	667 ± 18.84 ^a^	37.10 ± 1.68 ^ab^	18.14 ± 0.44 ^b^
	N0W2	3.52 ± 0.19 ^b^	47.13 ± 1.34 ^gh^	6.42 ± 0.44 ^g^	0.19 ± 0.01 ^hi^	676 ± 14.19 ^a^	34.62 ± 1.37 ^b^	18.66 ± 0.86 ^ab^
	N0W3	3.59 ± 0.19 ^ab^	52.50 ± 1.12 ^ef^	6.81 ± 0.18 ^fg^	0.22 ± 0.01 ^gh^	695 ± 15.07 ^a^	34.19 ± 1.61 ^b^	18.91 ± 0.60 ^ab^
	N1W1	3.81 ± 0.14 ^ab^	50.67 ± 2.26 ^fg^	9.53 ± 0.12 ^bc^	0.20 ± 0.01 ^hi^	672 ± 16.52 ^a^	37.93 ± 1.11 ^ab^	20.36 ± 0.63 ^ab^
	N1W2	3.97 ± 0.15 ^ab^	59.96 ± 1.56 ^d^	9.08 ± 0.37 ^cd^	0.27 ± 0.01 ^ef^	691 ± 22.05 ^a^	35.07 ± 1.35 ^ab^	21.02 ± 0.39 ^ab^
	N1W3	3.63 ± 0.17 ^ab^	65.25 ± 1.74 ^bc^	8.14 ± 0.15 ^de^	0.28 ± 0.01 ^def^	705 ± 17.27 ^a^	35.45 ± 1.86 ^ab^	19.28 ± 0.89 ^ab^
	N2W1	3.72 ± 0.19 ^ab^	57.12 ± 1.73 ^de^	10.72 ± 0.39 ^a^	0.35 ± 0.02 ^b^	681 ± 11.40 ^a^	38.71 ± 1.55 ^ab^	19.95 ± 1.66 ^ab^
	N2W2	3.84 ± 0.16 ^ab^	67.03 ± 1.91 ^ab^	10.46 ± 0.40 ^ab^	0.39 ± 0.01 ^a^	701 ± 18.21 ^a^	36.28 ± 0.86 ^ab^	21.41 ± 0.84 ^ab^
	N2W3	3.68 ± 0.10 ^ab^	70.84 ± 1.63 ^a^	7.71 ± 0.28 ^ef^	0.30 ± 0.02 ^cde^	718 ± 23.42 ^a^	37.52 ± 1.17 ^ab^	22.17 ± 0.84 ^a^
	N3W1	3.91 ± 0.18 ^ab^	54.18 ± 1.74 ^ef^	7.47 ± 0.18 ^ef^	0.25 ± 0.01 ^fg^	688 ± 19.94 ^a^	40.04 ± 2.16 ^a^	19.64 ± 0.80 ^ab^
	N3W2	3.72 ± 0.19 ^ab^	61.71 ± 1.10 ^cd^	9.85 ± 0.43 ^abc^	0.33 ± 0.01 ^bc^	708 ± 12.45 ^a^	36.67 ± 1.04 ^ab^	21.83 ± 1.14 ^a^
	N3W3	4.11 ± 0.19 ^a^	68.29 ± 1.87 ^ab^	7.03 ± 0.21 ^fg^	0.31 ± 0.01 ^cd^	715 ± 14.09 ^a^	38.15 ± 1.77 ^ab^	20.72 ± 2.00 ^ab^
	W	ns	***	***	***	ns	ns	ns
	N	ns	***	***	***	ns	ns	*
	W × N	ns	ns	***	*	ns	ns	ns
ANOVA								
Year (Y)		ns	ns	**	***	ns	ns	*
Water (W)		**	***	***	***	**	*	***
Nitrogen (N)		*	***	***	***	ns	ns	ns
Y × W		ns	ns	ns	ns	ns	ns	ns
Y × N		ns	ns	ns	ns	ns	ns	ns
W × N		ns	***	***	***	ns	*	ns
Y × W × N		ns	ns	ns	ns	ns	ns	ns

Note: Different lowercase letters in the same column indicate significant differences among the different treatments (*p* < 0.05). The *, **, and *** indicate significant differences among the different treatments at the levels of *p* < 0.05, *p* < 0.01, and *p* < 0.001, respectively. The ns means not significant at the level of *p* ≥ 0.05.

**Table 7 plants-14-02050-t007:** A comprehensive evaluation of water–nitrogen interaction effects on maize in the strip intercropping system and their ranking.

Treatment	*C* _11_	*C* _12_	*C* _13_	*C* _21_	*C* _22_	*C* _23_	*C* _24_	*C* _25_	*C* _31_
N1W1	0.217	0.240	0.274	0.254	0.290	0.293	0.308	0.162	0.331
N1W2	0.301	0.292	0.303	0.313	0.326	0.304	0.427	0.373	0.345
N1W3	0.319	0.323	0.322	0.364	0.296	0.258	0.451	0.339	0.318
N2W1	0.249	0.275	0.290	0.259	0.325	0.336	0.235	0.193	0.325
N2W2	0.338	0.359	0.348	0.325	0.351	0.341	0.319	0.345	0.337
N2W3	0.422	0.400	0.385	0.395	0.361	0.341	0.398	0.500	0.321
N3W1	0.271	0.308	0.312	0.292	0.314	0.365	0.192	0.188	0.339
N3W2	0.404	0.376	0.367	0.361	0.379	0.408	0.286	0.391	0.329
N3W3	0.412	0.389	0.379	0.401	0.347	0.332	0.291	0.354	0.353
$S^{+}$	0.422	0.400	0.385	0.401	0.379	0.408	0.451	0.500	0.353
$S^{-}$	0.217	0.240	0.274	0.254	0.290	0.258	0.192	0.162	0.318
Treatment	C_32_	C_33_	C_34_	C_35_	D^+^	D^-^	C	Sequence	
N1W1	0.273	0.352	0.235	0.322	0.163	0.033	0.167	9	
N1W2	0.326	0.334	0.300	0.329	0.094	0.091	0.491	6	
N1W3	0.348	0.306	0.310	0.337	0.090	0.100	0.525	5	
N2W1	0.309	0.395	0.380	0.325	0.138	0.056	0.288	7	
N2W2	0.357	0.384	0.412	0.335	0.069	0.112	0.620	4	
N2W3	0.380	0.292	0.337	0.342	0.035	0.160	0.819	1	
N3W1	0.291	0.282	0.278	0.328	0.136	0.051	0.272	8	
N3W2	0.334	0.364	0.364	0.338	0.052	0.136	0.725	2	
N3W3	0.367	0.267	0.348	0.343	0.063	0.134	0.681	3	
$S^{+}$	0.380	0.395	0.412	0.343	—	—	—	—	
$S^{-}$	0.273	0.267	0.235	0.322	—	—	—	—	

**Table 8 plants-14-02050-t008:** A comprehensive evaluation of water–nitrogen interaction effects on soybean in the strip intercropping system and their ranking.

Treatment	*C* _11_	*C* _12_	*C* _13_	*C* _21_	*C* _22_	*C* _23_	*C* _24_	*C* _25_	*C* _31_	*C* _32_	D^+^	D^−^	C	Sequence
N1W1	0.199	0.208	0.154	0.201	0.327	0.269	0.286	0.169	0.336	0.328	0.195	0.032	0.140	9
N1W2	0.295	0.287	0.243	0.302	0.322	0.299	0.423	0.394	0.321	0.338	0.115	0.106	0.480	6
N1W3	0.300	0.332	0.321	0.388	0.255	0.243	0.431	0.275	0.322	0.319	0.110	0.115	0.511	5
N2W1	0.251	0.251	0.193	0.224	0.372	0.339	0.240	0.230	0.342	0.326	0.163	0.056	0.256	8
N2W2	0.360	0.357	0.361	0.335	0.350	0.363	0.344	0.409	0.324	0.343	0.063	0.143	0.693	4
N2W3	0.430	0.418	0.462	0.410	0.344	0.347	0.411	0.476	0.333	0.346	0.019	0.197	0.910	1
N3W1	0.280	0.316	0.275	0.265	0.344	0.377	0.201	0.222	0.352	0.320	0.137	0.084	0.381	7
N3W2	0.395	0.377	0.403	0.360	0.361	0.398	0.283	0.366	0.331	0.344	0.059	0.157	0.726	3
N3W3	0.414	0.395	0.442	0.432	0.311	0.334	0.297	0.331	0.338	0.333	0.059	0.167	0.738	2
$S^{+}$	0.430	0.418	0.462	0.432	0.372	0.398	0.431	0.476	0.352	0.346	—	—	—	—
$S^{-}$	0.199	0.208	0.154	0.201	0.255	0.243	0.201	0.169	0.321	0.319	—	—	—	—

## Data Availability

The original contributions presented in this study are included in the article, and further inquiries can be directed to the corresponding author.

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
