# Peer review of "Rational Water and Nitrogen Regulation Can Improve Yield and Water–Nitrogen Productivity of the Maize (Zea mays L.)–Soybean (Glycine max L. Merr.) Strip Intercropping System in the China Hexi Oasis Irrigation Area"

_plants, 2025, doi:10.3390/plants14132050_

Round 1

Reviewer 1 Report (Previous Reviewer 2)

Comments and Suggestions for Authors

The article seems fairly well written except it is overly long. The results should be of interest to scientists and policymakers seeking optimum crop production with the least negative environmental impacts. 

Author Response

Response to Reviewers

Reply to Reviewer #1: The article seems fairly well written except it is overly long. The results should be of interest to scientists and policymakers seeking optimum crop production with the least negative environmental impacts.

Dear Reviewer,

Firstly, Thank you very much for your time involved in reviewing the manuscript, secondly, thank you for your respect for our efforts and your recognition of the contribution "Plants".

Responds to reviewer’s comments:

Thank you for your nice comments on our paper. According to your suggestions, we have reorganized the content of the full text, and tried our best to deleted unnecessary expressions, and merged some parts.

Please refer to the attached revised manuscript for details of the revisions.

Sincerely,

The Authors

Reviewer 2 Report (New Reviewer)

Comments and Suggestions for Authors

The manuscript demonstrates a relevant and timely study that addresses sustainable agricultural intensification in Hexi Oasis Irrigation Area in China. The manuscript needs to be concise because the reader will fail to concentrate on such a descriptive story. Some concerns need to be clarified, which are mentioned below:

Abstract: The author needs to rewrite the abstract concisely, mentioning only the key findings.

Introduction:

The author clearly explains the research gap and provides proper justification.

Materials and Methods:

Line 107-1108: On what basis did you select the different doses of nitrogen fertilizer that were considered as a treatment?

Line 131: You mention the unit hm-2. What is the actual meaning of it? Why don’t you use per hectare (ha-1) basis.

Results:

Did you check the relationship of yield and its related traits due to the water use and different doses of N fertilizer application in soybean and maize intercropping, due to the nitrogen fixation by the symbiosis process?

Discussion:

The author explains their obtained results properly in this section. However, please revise this section with the more supportive findings of the previous researchers.

 Overall comments: The manuscript needs revision.

Author Response

Response to Reviewers

Reply to Reviewer #2: The manuscript demonstrates a relevant and timely study that addresses sustainable agricultural intensification in Hexi Oasis Irrigation Area in China. The manuscript needs to be concise because the reader will fail to concentrate on such a descriptive story. Some concerns need to be clarified, which are mentioned below:

Dear Reviewer,

We feel great thanks for your professional review work on our article. According to your constructive suggestions, we have made extensive corrections to our previous draft, the detailed corrections are listed below.

Responds to reviewer’s comments:

Comment 1:Abstract: The author needs to rewrite the abstract concisely, mentioning only the key findings.

Response 1: Thanks for your suggestion. We have rewritten the abstract based on the revised manuscript.(L12-32)

Comment 2:Introduction:The author clearly explains the research gap and provides proper justification.

Response 2: Thank you for your recognition of introduction.

Comment 3:Materials and Methods:Line 107-1108: On what basis did you select the different doses of nitrogen fertilizer that were considered as a treatment?

Response 3: Thank you for pointing this out. The basis of the dosage of low nitrogen N1, medium nitrogen N2, high nitrogen N3 were according to local recommended fertilization level, set to its 60%, 80%, 100%.

Comment 4:Line 131: You mention the unit hm-2. What is the actual meaning of it? Why don’t you use per hectare (ha-1) basis.

Response 4: Thanks for your suggestion. We have changed “hm-2” into “ha-1” of the whole manuscript file.

Comment 5:Results:Did you check the relationship of yield and its related traits due to the water use and different doses of N fertilizer application in soybean and maize intercropping, due to the nitrogen fixation by the symbiosis process?

Response 5: In this experiment, the irrigation and fertilization systems for maize and soybeans were independent. Therefore, this paper did not analyze the relationship of yield and its related traits due to the water use and different doses of N fertilizer application in maize-soybean strip intercropping system. We are very grateful to you for providing us with such a good research idea. In the subsequent research, we will conduct in-depth research and analysis in this aspect.

Comment 6:Discussion:The author explains their obtained results properly in this section. However, please revise this section with the more supportive findings of the previous researchers.

Response 6: Thanks for your suggestion. We feel sorry for our poor writings. We have conducted a more in-depth analysis of the results of this study with compared the previous researchers.  (L443-572)

Please refer to the attached revised manuscript for details of the revisions.

Sincerely,

The Authors

Reviewer 3 Report (New Reviewer)

Comments and Suggestions for Authors
  1. Formatting Issue: The manuscript does not follow the journal's specific template. It is recommended to reformat the manuscript according to the Author Guidelines.
  2. Abstract: It is suggested to add 1-2 sentences briefly describing the research background. The results section should be more concise, focusing only on the key findings and critical data.
  3. Introduction: The introduction lacks a clear elaboration of the research background concerning the maize-soybean strip intercropping system and the theoretical significance of studying water-nitrogen interaction mechanisms. It is recommended to supplement this with a review of similar domestic and international studies.
  4. Introduction Weakness - Novelty: The expression of the study's innovation needs strengthening. The unique research perspective, such as investigating the "threshold effects of water-nitrogen coupling," is not sufficiently highlighted.
  5. Data Verification - Planting Density: The stated soybean planting density (row spacing: 0.3m, plant spacing: 0.17m, density: 90,909 plants hm⁻²) appears questionable. Please recalculate the density. Similarly, verify the maize planting density calculation.
  6. Methodology Detail: Key parameters for measurement methods (e.g., hydrolysis temperature for acid hydrolysis-DNS method, chromogenic agent concentration for semi-micro Kjeldahl method) are missing. Please supplement these critical experimental details.
  7. Discussion Depth - Comparisons & Applicability: The discussion lacks comparative analysis with water-nitrogen responses in other intercropping systems (e.g., maize-wheat), limiting the generalizability of the conclusions. The promotion value of the optimal water-nitrogen combination (N2W3) needs discussion regarding its regional climatic adaptability (e.g., threshold differences in arid/semi-arid vs. other regions).
  8. Discussion Depth - Result Discrepancies: Lines 517–519 note discrepancies between this study's results and those of Qu et al. and Wang et al., but fail to delve into potential reasons (e.g., differences in cropping systems, environmental conditions). This analysis should be expanded.
  9. Discussion Depth - Regional Specificity: Emphasize the differences in water-nitrogen response between the Hexi Corridor oasis (annual precipitation ~200 mm) and semi-humid regions, linking back to the "regionalization principle" mentioned in the introduction.
  10. Conclusion Clarity - Mechanism: While the conclusion identifies N2W3 as optimal, it does not explain why higher water and nitrogen (N3W3) negatively impact quality (e.g., reduced lysine in maize, crude fat in soybean). Supplement this with mechanistic analysis from physiological or resource competition perspectives.
  11. References: Citations within the manuscript text require superscript formatting. The reference list formatting also needs correction to comply with journal style.
Comments on the Quality of English Language

Read this article carefully again and correct any grammar errors that exist.

Author Response

Response to Reviewers

Reply to Reviewer #3:

Dear Reviewer,

We really appreciate you for the helpful comments to improve our manuscript. According to your constructive suggestions, we have made extensive corrections to our previous draft, the detailed corrections are listed below.

Responds to reviewer’s comments:

Comment 1: Formatting Issue: The manuscript does not follow the journal's specific template. It is recommended to reformat the manuscript according to the Author Guidelines.

Response 1: Your suggestion really means a lot to us. According to your request, we have recommended to reformat the manuscript according to the Author Guidelines.  

Comment 2: Abstract: It is suggested to add 1-2 sentences briefly describing the research background. The results section should be more concise, focusing only on the key findings and critical data.

Response 2: In accordance with your comments, we have added sentences briefly describing the research background (L12-14). And deleted unnecessary expressions of the results section, retained the key findings and critical data.

Comment 3: Introduction: The introduction lacks a clear elaboration of the research background concerning the maize-soybean strip intercropping system and the theoretical significance of studying water-nitrogen interaction mechanisms. It is recommended to supplement this with a review of similar domestic and international studies.

Response 3: Thank you for your reminder. We have rewritten the introduction, supplemented the research background concerning the maize-soybean strip intercropping system (L46-57) and the theoretical significance of studying water-nitrogen interaction mechanisms (L71-74).

Comment 4: Introduction Weakness - Novelty: The expression of the study's innovation needs strengthening. The unique research perspective, such as investigating the "threshold effects of water-nitrogen coupling," is not sufficiently highlighted.

Response 4: Thank you for your reminder. We have rewritten the introduction, supplemented the study's innovation (L76-82).

Comment 5: Data Verification - Planting Density: The stated soybean planting density (row spacing: 0.3m, plant spacing: 0.17m, density: 90,909 plants hm-2) appears questionable. Please recalculate the density. Similarly, verify the maize planting density calculation.

Response 5: We were really sorry for our careless mistakes. Thank you for pointing this out. We have recalculate the maize and soybean density, the soybean planting spacing is 0.10m.(L144)

Comment 6: Methodology Detail: Key parameters for measurement methods (e.g., hydrolysis temperature for acid hydrolysis-DNS method, chromogenic agent concentration for semi-micro Kjeldahl method) are missing. Please supplement these critical experimental details.

Response 6: Thank you for pointing this out, we have supplement these critical experimental details, so that other researchers can replicate our experiment well. (L195-213)

Comment 7: Discussion Depth - Comparisons & Applicability: The discussion lacks comparative analysis with water-nitrogen responses in other intercropping systems (e.g., maize-wheat), limiting the generalizability of the conclusions. The promotion value of the optimal water-nitrogen combination (N2W3) needs discussion regarding its regional climatic adaptability (e.g., threshold differences in arid/semi-arid vs. other regions).

Response 7: Your suggestion really means a lot to us. We feel sorry for our poor writings. We have conducted a more in-depth analysis with water-nitrogen responses in other intercropping systems, and discussed the differences in water and nitrogen thresholds between the semi-arid area and this experimental area.(L458-465)

Comment 8: Discussion Depth-Result Discrepancies: Lines 517-519 note discrepancies between this study's results and those of Qu et al. and Wang et al., but fail to delve into potential reasons (e.g., differences in cropping systems, environmental conditions). This analysis should be expanded.

Response 8:Thank you very much for your comments and suggestions. We have conducted an in-depth analysis of the reasons for the differences from the results of Qu et al. and Wang et al.(L545-554)

Comment 9: Discussion Depth - Regional Specificity: Emphasize the differences in water-nitrogen response between the Hexi Corridor oasis (annual precipitation ~200 mm) and semi-humid regions, linking back to the "regionalization principle" mentioned in the introduction.

Response 9: Thank you for pointing this out. We have analyzed the differences in water-nitrogen response between the Hexi Corridor oasis and semi-humid regions.(L508-516)

Comment 10: Conclusion Clarity - Mechanism: While the conclusion identifies N2W3 as optimal, it does not explain why higher water and nitrogen (N3W3) negatively impact quality (e.g., reduced lysine in maize, crude fat in soybean). Supplement this with mechanistic analysis from physiological or resource competition perspectives.

Response 10: In accordance with your comments, we have added the mechanistic analysis from physiological and resource competition perspectives to explain why higher water and nitrogen (N3W3) negatively impact quality.(L581-584)

Comment 11: References: Citations within the manuscript text require superscript formatting. The reference list formatting also needs correction to comply with journal style.

Response 11: Your suggestion really means a lot to us. According to your request, we have modified the reference list formatting, but the references in the latest published paper in Plants were not superscript formatting, so we retained the original format.

Comment 12: Read this article carefully again and correct any grammar errors that exist.

Response 12: Thank you very much for your comments and suggestions. We have revised strictly in accordance with the comments, combed the full text, checked sentence structure and English grammar, and made professional linguistic touch-ups.

Please refer to the attached revised manuscript for details of the revisions.

Sincerely,

The Authors

Round 2

Reviewer 3 Report (New Reviewer)

Comments and Suggestions for Authors

The author provided a good answer to the reviewer's question, but the manuscript still did not follow the journal guidelines for formatting.

Author Response

Response to Reviewers

Reply to Reviewer:

Dear Reviewer,

We really appreciate you for the helpful comments to improve our manuscript. According to your constructive suggestions, we have revised our manuscript follow the journal guidelines for formatting. Meanwhile,we have combed the full text, checked sentence structure and English grammar, and made professional linguistic touch-ups. Please refer to the attached revised manuscript for details of the revisions.

Sincerely,

The Authors

This manuscript is a resubmission of an earlier submission. The following is a list of the peer review reports and author responses from that submission.

Round 1

Reviewer 1 Report

Comments and Suggestions for Authors

The publication submitted for evaluation is devoted to the problem of intercropping of very important plants such as maize and soybean, which is important from the production and environmental point of view. The research was carried out correctly, the results were subjected to exhaustive statistical analysis and discussion, and the conclusions were correctly drawn. I rate the work highly in terms of content, language and editing. We assess the strengths and weaknesses of the study to make a recommendation for further action. Based on the strengths and weaknesses identified, we recommend that the article undergo a revision.These revisions will improve the article's potential for publication and its contribution to the field of sustainable agriculture.

Comment 1: The abstract is lengthy and could be shortened to provide a more concise summary of the research.

Comment 2: It should explicitly state what sets it apart from earlier works and highlight the novelty of its contributions.

Comment 3: Line 187-193: Please include a reference for the formula for WP, IP, NPF and NAE.

Comment 4: Line 198: Please specify the details i.e. brand, serial number, company, and location of the company for the scale.

Comment 5: Line 210: Please clarify where the various agar media were obtained. Please clarify how the microbial enumeration was done.

Comment 6: Figure 5, Figure 6: What do the stars mean? What do the error bars mean? How many replicates were used for the means? All this information should be included in the figure caption.

Comment 7:Restructure some sentences and phrases for better comprehension.

Author Response

Response to Reviewers

Reply to Reviewer #1: The publication submitted for evaluation is devoted to the problem of intercropping of very important plants such as maize and soybean, which is important from the production and environmental point of view. The research was carried out correctly, the results were subjected to exhaustive statistical analysis and discussion, and the conclusions were correctly drawn. I rate the work highly in terms of content, language and editing. We assess the strengths and weaknesses of the study to make a recommendation for further action. Based on the strengths and weaknesses identified, we recommend that the article undergo a revision.These revisions will improve the article's potential for publication and its contribution to the field of sustainable agriculture.

Dear Reviewer,

Firstly, Thank you very much for your time involved in reviewing the manuscript, secondly, thank you for your respect for our efforts and your recognition of the contribution "Plants".

Responds to reviewer’s comments:

Comment 1: The abstract is lengthy and could be shortened to provide a more concise summary of the research.

Response 1: In accordance with your comments, we have rewritten the abstract to basically meet the requirements.

Comment 2: It should explicitly state what sets it apart from earlier works and highlight the novelty of its contributions.

Response 2:Thank you for your reminder. We have rewritten the introduction, and have described in detail the novel aspects of this paper.

Comment 3: Line 187-193: Please include a reference for the formula for WP, IP, NPF and NAE.

Response 3: In accordance with your comments, we have added the reference for the formula for WP, IP, NPF and NAE.(L182-188)

Comment 4: Line 198: Please specify the details i.e. brand, serial number, company, and location of the company for the scale.

Response 4: Thanks for your suggestion. We have added the details, i.e., brand, serial number, company, and location of the company for the scale.(L194-195)

Comment 5: Line 210: Please clarify where the various agar media were obtained. Please clarify how the microbial enumeration was done.

Response 5: In accordance with your comments, we have supplemented the company information of obtain the various agar media, as well as the method of microbial enumeration.(L209-221)

Comment 6: Figure 5, Figure 6: What do the stars mean? What do the error bars mean? How many replicates were used for the means? All this information should be included in the figure caption.

Response 6: Thank you for pointing this out. Bars and error bars stand to represent averaged values ± standard errors (n = 3), the *, ** and *** indicate significant differences among different treatments at the levels of p < 0.05, p < 0.01 and p < 0.001. The ns means not significant at the level of p ≥ 0.05. We have supplemented this information in the figure caption.

Comment 7:Restructure some sentences and phrases for better comprehension.

Response 7: Thank you for your affirmation, we will continue to work hard. We have reorganized and written the introduction and conclusions. At the same time, the materials and methods, mesults and discussion was revised according to the requirements of the reviewers, and tried our best to improve the language expression of the full text.

Please refer to the attached revised manuscript for details of the revisions.

Sincerely,

The Authors

Reviewer 2 Report

Comments and Suggestions for Authors

Objectives included investigating different water-nitrogen combinations on multiple indicators of the maize-soybean strip intercropping system, including yield, water-nitrogen productivity, quality, and soil quality in the China Hexi Oasis Irrigation Area. Main results included the highest yield was with irrigation for maize and soybeans set at 100% ETc, with nitrogen application rates of 354.78~422.51 kg hm and 67.02~81.73 kg hm. The authors conclude that “this provides guidance for enhancing yield and quality in maize-soybean strip intercropping in the oasis agricultural area of the Hexi Corridor, achieving the dual objectives of high and stable yields and soil sustainability”. The article seems fairly well written except it is overly long. The results should be of interest to scientists and policymakers seeking optimum crop production with the least negative environmental impacts. A few minor suggestions follow.

.

ETc is first mentioned in the abstract. But it was not defined in the manuscript. A quick search suggests that ETc is a common abbreviation for crop evapotranspiration, but it is usually defined in the abstract of published studies.

The manuscript states “…nitrogen application rate was the dominant factor influencing soybean yield, with irrigation amount playing a secondary role”. In the US North Central soybean region, soybean N fertilization is usually not recommended (see example links below). Perhaps consider a few more words in Intro and Discussion sections about nuances of N fertilization response with soybean. A brief literature review suggests that Liao et al. (2022) reported that N fertilization improved soybean yield in China.

Liao et al., 2022 https://www.sciencedirect.com/science/article/pii/S0378377422002359

https://extension.illinois.edu/sites/default/files/2025-03/illinois-soybean-management.pdf#:~:text=Soybeans%20take%20up%20~5%20lbs,N%20per%20acre.

https://extension.umn.edu/crop-specific-needs/soybean-fertilizer-guidelines#nitrogen-1078560

This may be the longest Abstract I have seen. Most abstract recommendations are less than 300 words. The current abstract is over 600 words. Consider reducing the abstract to only the most essential items.  

The article is very long, perhaps the longest scientific journal article I’ve reviewed or read in the last few years. Consider reducing the reported study to only the most essential items.  

Author Response

Response to Reviewers

Reply to Reviewer #2: Objectives included investigating different water-nitrogen combinations on multiple indicators of the maize-soybean strip intercropping system, including yield, water-nitrogen productivity, quality, and soil quality in the China Hexi Oasis Irrigation Area. Main results included the highest yield was with irrigation for maize and soybeans set at 100% ETc, with nitrogen application rates of 354.78~422.51 kg hm and 67.02~81.73 kg hm. The authors conclude that “this provides guidance for enhancing yield and quality in maize-soybean strip intercropping in the oasis agricultural area of the Hexi Corridor, achieving the dual objectives of high and stable yields and soil sustainability”. The article seems fairly well written except it is overly long. The results should be of interest to scientists and policymakers seeking optimum crop production with the least negative environmental impacts. A few minor suggestions follow..

Dear Reviewer,

Firstly, Thank you very much for your time involved in reviewing the manuscript, secondly, thank you for your respect for our efforts and your recognition of the contribution "Plants".

Responds to reviewer’s comments:

Comment 1: ETc is first mentioned in the abstract. But it was not defined in the manuscript. A quick search suggests that ETc is a common abbreviation for crop evapotranspiration, but it is usually defined in the abstract of published studies.

Response 1: Thank you for your reminder. We have defined ETc in the abstract.(L33)

Comment 2: The manuscript states “…nitrogen application rate was the dominant factor influencing soybean yield, with irrigation amount playing a secondary role”. In the US North Central soybean region, soybean N fertilization is usually not recommended (see example links below). Perhaps consider a few more words in Intro and Discussion sections about nuances of N fertilization response with soybean. A brief literature review suggests that Liao et al. (2022) reported that N fertilization improved soybean yield in China.

Liao et al., 2022 https://www.sciencedirect.com/science/article/pii/S0378377422002359

https://extension.illinois.edu/sites/default/files/2025-03/illinois-soybean-management.pdf#:~:text=Soybeans%20take%20up%20~5%20lbs,N%20per%20acre.

https://extension.umn.edu/crop-specific-needs/soybean-fertilizer-guidelines#nitrogen-1078560.

Response 2: Thank you for your reminder. We have carefully read the literature you recommended and quoted it in the introduction. The effects of nitrogen application on soybeans may vary significantly depending on regional climatic conditions, soil fertility, and cultivation management measures. In Minnesota, soybeans can meet their nitrogen requirements through rhizobial nitrogen fixation and soil mineralization, making the yield-increasing effect of additional nitrogen application limited. In contrast, in semi-arid and semi-humid regions of China, rational increases in nitrogen application can enhance yield and water use efficiency by improving leaf photosynthetic performance. Therefore, the management of nitrogen for soybeans should adhere the regionalization principle, with fertilization strategies tailored to the specific characteristics of the local soil-climate-crop system to achieve synergistic improvements in both yield and nitrogen use efficiency.(L70-77)

Comment 3: This may be the longest Abstract I have seen. Most abstract recommendations are less than 300 words. The current abstract is over 600 words. Consider reducing the abstract to only the most essential items.

Response 3: In accordance with your comments, we have rewritten the abstract to basically meet the requirements.

Comment 4:The article is very long, perhaps the longest scientific journal article I’ve reviewed or read in the last few years. Consider reducing the reported study to only the most essential items.  

Response 4: In accordance with your comments, We have simplified the content of the paper, but in order to ensure the integrity and integrity of the paper, the scope of simplification is limited, please forgive me.

Please refer to the attached revised manuscript for details of the revisions.

Sincerely,

The Authors

Round 2

Reviewer 1 Report

Comments and Suggestions for Authors

NO

Comments on the Quality of English Language

NO